# Auditory stimulation during REM sleep modulates REM electrophysiology and cognitive performance

Miguel Navarrete [1,2✉], Viviana Greco[1], Martyna Rakowska[1], Michele Bellesi [3] & Penelope A. Lewis [1✉]

REM sleep is critical for memory, emotion, and cognition. Manipulating brain activity during REM could improve our understanding of its function and benefits. Earlier studies have suggested that auditory stimulation in REM might modulate REM time and reduce rapid eye movement density. Building on this, we studied the cognitive effects and electroencephalographic responses related to such stimulation. We used acoustic stimulation locked to eye movements during REM and compared two overnight conditions (stimulation and no-stimulation). We evaluated the impact of this stimulation on REM sleep duration and electrophysiology, as well as two REM-sensitive memory tasks: visual discrimination and mirror tracing. Our results show that this auditory stimulation in REM decreases the rapid eye movements that characterize REM sleep and improves performance on the visual task but is detrimental to the mirror tracing task. We also observed increased beta-band activity and decreased theta-band activity following stimulation. Interestingly, these spectral changes were associated with changes in behavioural performance. These results show that acoustic stimulation can modulate REM sleep and suggest that different memory processes underpin its divergent impacts on cognitive performance.

[1] Cardiff University Brain Research Imaging Centre (CUBRIC), School of Psychology, Cardiff University, Maindy Rd, Cardiff CF24 4HQ, UK. [2] Psychology and Biobehavioral Sciences Department, St. Jude Children's Research Hospital, Memphis, TN 38105, USA. [3] School of Biosciences and Veterinary Medicine, University of Camerino, Via Gentile III Da Varano, 62032 Camerino (MC), Italy. ✉email: mnavarretem@gmail.com; LewisP8@cardiff.ac.uk

Rapid eye movement (REM) sleep is believed to be important for memory consolidation[1-3] and has been shown to be involved in the regulation of complex conceptual memories[4,5] as well as emotional memories[6], visual[7], and motor procedural memories[7,8]. The up-regulation of gene expressions during REM may facilitate long-term potentiation in the hippocampus[9], as well as reorganization of synaptic plasticity in the cortex[10-12]. If we can develop a method for manipulating REM, this will provide an invaluable tool for gaining a better understanding of the various roles of this sleep stage in memory and cognition.

A series of studies from the '1980s and '1990s provided initial evidence that it is possible to alter electrophysiological characteristics of REM through sensory stimulation. These early studies suggested that brief auditory stimulation triggered by REM Eye Movements (EMs) can increase the length of REM sleep in humans[13-15], while also reducing the density of EMs[13,14]. Importantly, a number of studies also managed to manipulate REM through auditory stimulation that was not locked to EMs, e.g. refs. [16-18]. In most cases this was successful and increased REM duration while reducing EMs. However, a direct comparison of the EM-linked and non-EM-linked methods showed that the former was significantly more effective when it came to boosting memory consolidation for a Morse code comparison task[15]. To our knowledge, this was the only study which looked at the impact of such stimulation on memory consolidation. Although the result was very encouraging, the authors cautioned that because their REM stimulation used the same auditory modality as their morse code learning task, and it might have reinstated the task.

In the current study, we build on this literature by exploring how EM-triggered auditory stimulation actively alters REM electrophysiology, and how such stimulation impacts the consolidation of non-auditory REM-dependent memory tasks for which replay could not be triggered by the click stimuli. If that is the case, we predict auditory stimulation would lead to changes in EEG activity, which would in turn predict cognitive performance. To test this, we examined memory performance and the associated post-stimulus spectral EEG activity in two tasks that have been previously associated to REM related consolidation: the visual texture discrimination task (VDT), a task which has been shown to be sensitive to REM[7,19,20], and the mirror tracing task (MTT)[8] (see "Methods" section for details). Recent imaging and behavioural studies suggested that REM sleep helps to stabilize previous learning of the VDT, possibly by regulating synaptic activity during REM[7,21]. Post-sleep learning of the MTT has been shown to increase REM density[22], and an interesting model proposes that early consolidation of the MTT depends on REM sleep, while later consolidation (once the skill is already quite well learned) depends on NREM sleep[23]. Others have shown that high acetylcholine levels and associated increases in REM density during REM, triggered by an acetylcholinesterase inhibitor, facilitate enhanced sleep-dependent MTT gains[24,25].

Given the prior literature, we hypothesized that auditory stimulation triggered by EMs would lead to changes in REM length, REM eye movements (EMs) and EEG activity, which would, in turn, alter consolidation of our REM-sensitive memory tasks. Our findings revealed that auditory stimulation can indeed modulate both EEG and behavioural consolidation. Interestingly, our data also suggest that this stimulation method might enhance some memory processes while disturbing others.

## Results

We compared sleep structure, behavioural performance, and electrophysiology of sleep nights in conditions of no-stimulation control (CNT) and auditory stimulation (STM) (Fig. 1a). The behavioural tasks included: (i) an attentional task: the psychomotor vigilance test (PVT), (ii) a visual task: the texture discrimination test (VDT), and (iii) a procedural task: the mirror tracing test (MTT) (Supplementary Fig. 1). Using a semi-automated closed-loop system, short click sounds (50 ms pink noise) were applied locked to EMs during REM periods in the STM night. Our simulation setup was manually turned on in the presence of EMs during REM and turned off in the presence of any arousal or changes in the sleep stage. The same protocol was run but sounds were muted during the CNT night (Fig. 1b and Supplementary Fig. 2). This resulted in an average of $88.69 \pm 1.86\%$ of clicks applied during REM sleep, and $33.24 \pm 15.66\%$ of clicks applied on the EM that trigger the stimulus (Supplementary Table 1). Clicks in the STM night did not cause any changes in the sleep macrostructure, as seen in Table 1. Likewise, neither the proportions of tonic and phasic REM nor the duration of these REM sleep periods was modified. However, consistent with previous studies[13,15,26], auditory stimulation did cause a mean reduction of 1.8 EM/min (95% CI: 0.07–3.54 EM/min; $Z = -2.49$, $p = 0.013$, Wilcoxon signed-rank test, Table 1), representing a mean decrease of 15.15% (95% CI: 3.40–26.90%) in EM density. The increase of inter-EM interval did not quite pass significance (Supplementary Table 2, $t(18) = 2.03$, $p = 0.057$), but the probability of detecting an EM in a 2.5 s window after the click was significantly larger in the CNT compared to the STM condition ($t(18.0) = 2.76$, $p = 0.013$). Nevertheless, the total number of EMs detected during REM sleep was not altered by the stimulation ($t(18) = -1.09$, $p = 0.292$, Supplementary Table 2). As for the behavioural results, overnight performance gain (percentage of change in performance from the pre-sleep session to the post-sleep session[7]) on the PVT did not differ between conditions ($t(19.0) = 1.21$, $p = 0.240$, Fig. 1c), indicating that clicks did not disturb alertness after sleep. However, overnight performance improvement did differ between conditions for both visual and procedural tasks. We therefore focus on these two tasks for further analyses.

Visual learning, evaluated by the VDT, improved more over the STM night than over the CNT night. Specifically, we found an average difference of 20.71% (95%CI: 6.22–35.19%) in overnight performance gain on the visual task in STM compared with CNT ($t(17.0) = -3.02$; $p = 0.0078$, Fig. 1d). Using multivariable linear regression, we evaluated how this overnight performance gain depended on two variables: (i) the condition of stimulation (STM vs. CNT) and (ii) time spent in each sleep stage (i.e., % $gain_{VDT} = \beta_0 + \beta_1 (Condition) + \beta_2 (Sleep Stage)$). This showed no association between overnight performance gain and time spent in either NREM or REM sleep (both $p > 0.263$, Supplementary Table 3), with only condition of stimulation predicting performance gain (all $p < 0.014$, Supplementary Table 3). However, time in tonic REM and the condition of stimulation together predicted 25.5% of the variance (Model fit: $R^2 = 0.30$, $F(2,34) = 7.17$, $p = 0.002$). Thus, overnight performance gain on the VDT reduced with increased time in tonic REM ($\beta_2 = -0.66$, $p = 0.014$) when the condition of stimulation was considered (STM vs. CNT: $\beta_1 = 20.50$, $p = 0.003$, Fig. 1f, Supplementary Table 4). Total time in phasic REM did not predict performance gain on the visual task ($\beta_2 = 0.17$, $p = 0.315$), but Condition was a significant predictor when considering both phasic and tonic REM (see Supplementary Table 4).

In contrast to VDT learning, overnight performance gain on the MTT was negative over the STM night, by a mean of 11.60% (95% CI from −21.97 to −1.23), and significantly more negative than the performance gain over the CNT night ($t(19.0) = 2.34$; $p = 0.030$) (Fig. 1e). As in the VDT, multivariable linear regressions of the MTT (i.e., %$gain_{MTT} = \beta_0 + \beta_1 (Condition) + \beta_2$

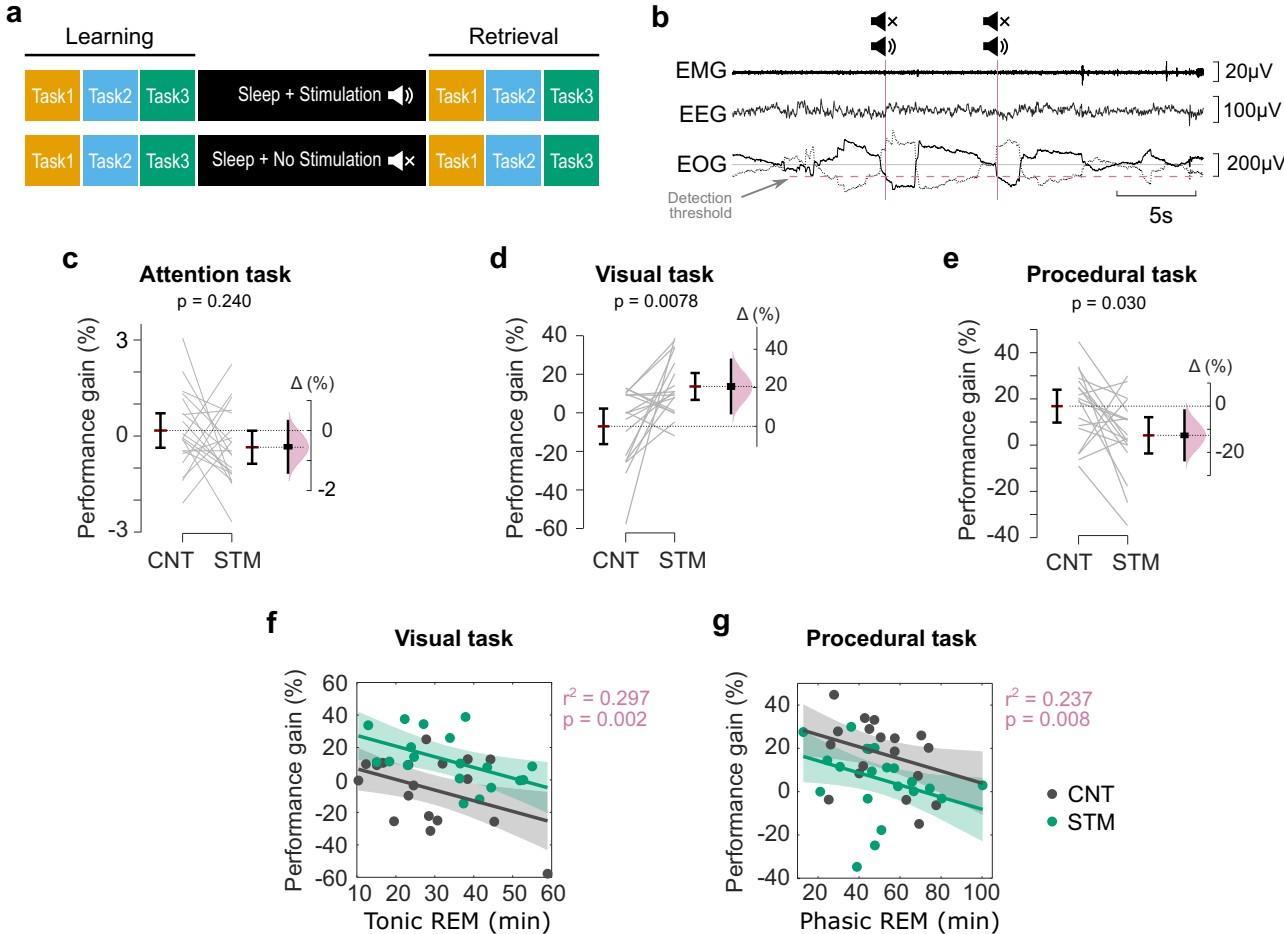

**Fig. 1 Study design and behavioural results. a** Participants learned three different cognitive tasks: (i) an attentional task ($n = 20$): the psychomotor vigilance test (PVT), (ii) a visual task ($n = 18$): the texture discrimination test (VDT), and (iii) a procedural task ($n = 20$): the mirror tracing test (MTT). Then, participants were allowed to sleep for ~7 h, and acoustic clicks locked to eye movements (EM) were applied during REM periods in the stimulation procedure (STM), or the sound was muted in the control procedure (CNT). After sleep, participants were tested on the same cognitive tasks. The order of the nights (CNT, STM) and the order of the tasks were randomized between participants, but the tasks were applied in the same order within participants for pre- and post-sleep tests. **b** Automatic detection of EM. One EOG channel was constantly monitored, and the EM detection was activated when in stable REM sleep. An amplitude threshold (50–100 µV) was manually set, and when the EOG amplitude crossed this level, an auditory click was applied, followed by a 2.5-s pause in the stimulation. Clicks were muted in the CNT condition. **c** No differences were evident in the overnight performance gain for the attention task (PVT). Changes in the overnight performance gain and the effect sizes for STM and CNT conditions in the visual (VDT) (**d**) and procedural (MTT) (**e**) tasks. Multivariable linear models suggest that the time spent in tonic REM predicts overnight changes in visual task performance (% gain = 13.5 + 20.5 C–0.66tR) (**f**), whereas the time spent in phasic REM predicts the performance gains on the procedural task (% gain = 31.9.5–12.0 C–0.28pR) (**g**). Error bars indicate mean ±95% CI and Δ visualizes effect sizes by the difference of means. The distribution curve for the effect size indicates the resampled distribution of Δ given the observed data. For the multivariate linear regressions in **f** and **g**: C = condition (CNT/STM), tR = tonic REM and pR = phasic REM, shading corresponds to 95% CI for the responses.

(Sleep Stage)) showed no overnight performance gain when only time spent in each sleep stage was included as a covariate (all NREM stages $p > 0.603$, and $p = 0.083$ for REM time, Supplementary Table 5). However, when including both stimulation condition and time in phasic REM as predictors, the multivariate regression gave a significant linear fit (Model fit: $R^2 = 0.24$, $F(2,36) = 5.6$, $p = 0.0076$). Thus, the combination of time in phasic REM and condition of stimulation predicted the extent of negative performance gain for the MTT (time in phasic REM: $\beta_2 = -0.28$, $p = 0.036$; STM vs. CNT: $\beta_1 = -12.04$, $p = 0.016$, Fig. 1g, Supplementary Table 6). Notably, however, neither time in tonic REM ($\beta_2 = -0.03$, $p = 0.896$) nor the proportion of phasic REM within total REM ($\beta_2 = -0.29$, p = 0.196) predicted overnight performance gain for this task when the condition of stimulation was not included in the model. On the other hand,

the condition of stimulation was still a significant predictor in both models (Supplementary Table 6).

These results show that time spent in tonic and phasic REM predicts overnight performance gain when the condition of stimulation is considered. Next, we were interested in determining whether differences between the time spent in tonic or phasic REM also predict differences in performance between STM and CNT nights. We, therefore, evaluated whether between-condition (CNT-vs.-STM) differences in tonic/phasic REM time correlated with between-condition differences in performance gain for VDT or MTT, depending on their significant predictor. This showed a correlation between CNT-vs.-STM differences in time spent in tonic REM and CNT-vs-STM differences in performance gain on the VDT (Pearson's $\rho = -0.5$, $p = 0.043$). However, between-condition differences in phasic REM did not correlate with

**Table 1 Sleep characteristics and differences between experimental nights.**

| Variables | CNT | | STM | | Cohen's d | p |
|---|---|---|---|---|---|---|
| | *M* | SD | *M* | SD | | |
| *Sleep* | | | | | | |
| Recording (min) | 447.03 | 60.29 | 463.70 | 44.61 | −0.32 | 0.188 |
| TST (min) | 433.18 | 57.41 | 453.73 | 46.64 | −0.39 | 0.100 |
| N1 (%) | 10.34 | 7.44 | 8.45 | 5.00 | 0.30 | 0.295 |
| N2 (%) | 41.10 | 8.39 | 43.25 | 7.64 | −0.27 | 0.141 |
| N3 (%) | 27.10 | 10.35 | 27.86 | 9.52 | −0.08 | 0.872 |
| NREM (%) | 78.54 | 4.47 | 79.56 | 4.89 | −0.22 | 0.541 |
| REM (%) | 18.34 | 4.40 | 18.25 | 4.63 | 0.02 | 0.991 |
| Latency (min) | 13.19 | 12.74 | 12.84 | 9.29 | 0.03 | 0.905 |
| WASO (min) | 13.84 | 16.28 | 9.98 | 12.97 | 0.26 | 0.538 |
| Efficiency (%) | 91.78 | 11.52 | 94.39 | 5.36 | −0.29 | 0.243 |
| Arousals | 71.11 | 49.41 | 58.25 | 43.70 | 0.28 | 0.758 |
| *NREM* | | | | | | |
| SO density (SO/min) | 10.96 | 4.81 | 11.67 | 4.93 | −0.15 | 0.323 |
| SP density (SP/min) | 5.16 | 0.86 | 5.53 | 0.54 | −0.52 | 0.107 |
| *REM* | | | | | | |
| Phasic REM (%) | 11.61 | 3.48 | 10.98 | 4.32 | 0.16 | 0.581 |
| Tonic REM (%) | 6.73 | 2.59 | 7.27 | 2.44 | −0.22 | 0.629 |
| EM density (EM/min) | 6.86 | 3.91 | 5.07 | 1.43 | 0.62 | **0.013** |
| Number of REM segments | 17.20 | 9.19 | 20.60 | 8.61 | −0.38 | 0.117 |

$N = 19$ subjects for each condition. All percentages are based on TST. NREM includes stages N2 and N3 only.
*TST* total sleep time, *SO* slow oscillations at Fz, *SP* spindles at Cz, *EM* eye movements during REM sleep, *WASO* wake after the sleep onset.

between-condition differences in performance gain on the MTT (Pearson's $\rho = −0.18$, $p = 0.453$) when calculated in the same way. These data suggest that while stimulation significantly impacted on the MTT performance, this is not due to between-condition differences in the time spent in tonic or phasic REM alone.

Overall, our behavioural results show that auditory stimulation of REM can affect overnight performance on both the VDT and MTT. The percentage of stimulated EMs was correlated with the overnight change in performance for MTT (Pearson's $\rho = −0.54$, $p_{uncorr} = 0.016$) but not for VDT (Pearson's $\rho = 0.10$, $p_{uncorr} = 0.695$). However, MTT performance was not correlated with the total number of clicks applied during REM sleep (Pearson's $\rho = −0.43$, $p_{uncorr} = 0.067$) (Supplementary Table 7). Also, because our three tasks were scheduled in a row, they could potentially have interfered with each other. However, we found no correlation in the overnight performance gain between VDT and MTT (Pearson's $\rho = −0.30$, $p_{corr} = 0.468$), between MTT and PVT (Pearson's $\rho = −0.11$, $p_{corr} = 0.483$), or between VDT and PVT (Pearson's $\rho = −0.17$, $p_{corr} = 0.483$). Similarly, we found no correlations between performance gain on any task and mood (all $p_{corr} > 0.05$, Supplementary Table 8), indicating that neither the applied tasks nor the stimulation interfered with the emotional state of the participants. Additionally, a two-way ANOVA was used to determine the effect of Condition (CNT vs. STM) and Learning Order (Night 1 vs Night 2) on the performance of each task. For the VDT this ANOVA revealed that there was not a significant interaction between the effects of Condition and Learning Order ($F(1, 32) = 2.31$, $p = 0.139$), and Simple Main Effects showed only Condition had a significant effect on VDT performance (Condition $p = 0.001$, Learning Order $p = 0.942$). A similar two-way ANOVA for MTT showed no interactions between the effects of Condition and Learning Order ($F(1, 36) = 1.08$, $p = 0.305$), and only Condition presented significant Simple Main Effects on MTT performance (Condition $p = 0.029$, Learning Order $p = 0.305$).

Moving to our electrophysiological analysis, we evaluated scalp EEG responses to auditory stimuli using scalp ERP and ERSP for all channels. Firstly, for the ERP we examined the incidence of N1 and P2 potentials which have previously been shown to be distinctive for different sleep stages[27]. Interestingly, P2 was delayed compared to the literature, peaking at 250–300 ms after the click (Fig. 2a) and visible in the raw data (Supplementary Fig. 3), but neither N1 nor P2 was significantly different between STM and CNT trials after FDR correction. Additionally, the ERP amplitudes averaged across all channels showed a significant decrease of around 500 ms after the click that was maintained for ~1 s (Fig. 2a, thick black bar). Secondly, in the mean ERSP across all scalp channels we found two main frequency clusters induced by the stimulus (Fig. 2b). Shortly after the P2 component of the ERP, an increase in beta frequency (cluster $\beta c$) was evident, lasting ~800 ms and peaking at ~600 ms after the stimulus onset. Additionally, a decrease in theta frequency (cluster $\theta c$) was detected from ~700 ms after the stimulus until around 1500 ms. Centro-frontal response of both beta and theta clusters (Supplementary Fig. 4) is in keeping with previous studies reporting induced potentials following auditory stimulation in REM sleep[28], and they are also evident when using threshold-free or stricter clusters (Supplementary Fig. 5). Although it is possible that the increase in beta marks microarousals triggered by the stimulus, we found no difference between STM and CNT in the EMG and EOG trials, which should highlight arousals (Supplementary Fig. 6). Thus, the occurrence of stimulation-induced arousals seems unlikely.

Finally, we wanted to determine whether these brain dynamics were related to changes in memory performance. To this end, we evaluated the linear association between the overnight performance gain on the behavioural tasks and both the negative ERP amplitude deflection and the time–frequency clusters across all electrodes in both conditions (STM and CNT). We found no linear associations between the ERP and performance for either VDT (all corrected $p > 0.882$) or the MTT (all corrected $p > 0.509$). However, we did find linear associations between beta and theta spectral EEG response and changes in performance. Specifically, after FDR correction for the number of scalp channels ($n = 21$), we found that VDT performance gain was

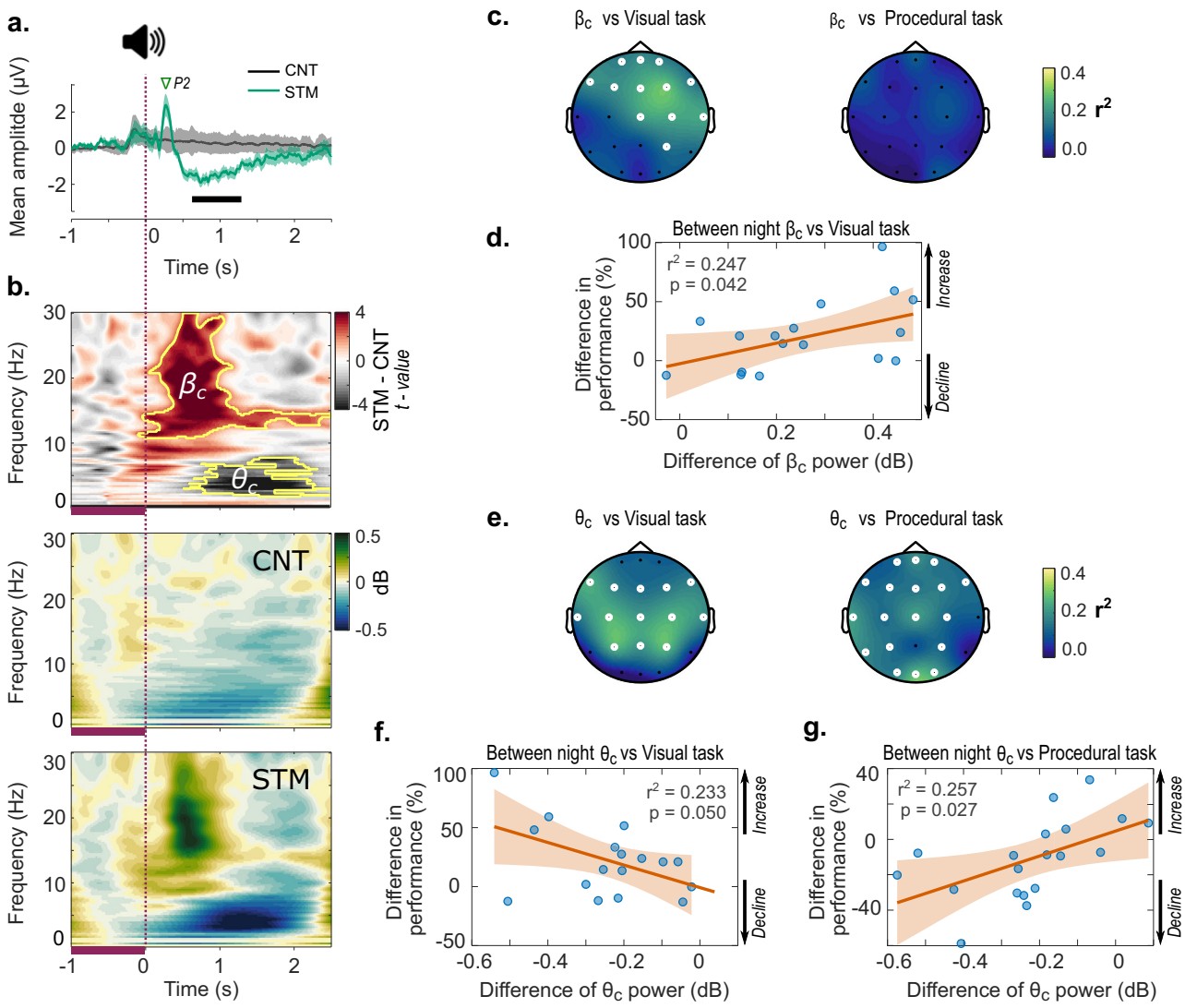

**Fig. 2 ERP and spectral response to auditory stimuli of REM. a** Event-related potential (ERP) averaged across all channels in response to auditory stimulus-locked to EM during REM. The thick black bar indicates significant differences between conditions. Shading indicates mean ±95% CI. **b** Mean event-related spectral perturbations (ERSP) across all channels in response to the auditory click. The analysis of spectral differences between conditions indicated two main clusters in beta ($\beta c$) and theta ($\theta c$) frequencies ($p < 0.05$, cluster-corrected). Darker purple bars between −1 and 0 s below the time-frequency plots indicate the baseline period for changes in frequency power (dB). Changes of power within $\beta c$ (**c**) and $\theta c$ (**e**) for each electrode were fit to linear regressions predicting overnight improvement in the performance of the visual (VDT) and procedural (MTT) tasks. White channels indicate significant linear fits after FDR correction ($q < 0.05$), indicating a linear relationship between behavioural performance and cluster power. Hence, between night differences in relative power (for significant channels) (CNT-vs.-STM) correlated with overnight performance gain for the visual task between conditions (**d**). Likewise, between-condition (CNT-vs-STM) differences in relative $\theta c$ power (for significant channels) marginally correlated with the decline in the overnight performance for the visual task (**f**) while the same difference in $\theta c$ power correlated with improvements in performance for the procedural task (**g**). Positive CNT-vs.-STM differences in performance indicate improvements in performance over the STM night, whereas negative values indicate a decline in performance over the STM night. Likewise, CNT-vs.-STM differences in power greater than 0 indicate larger power in STM, whereas changes in power lower than 0 indicate larger power in CNT. Shading corresponds to 95% CI of the response for regression plots in (**d**, **f**, and **g**).

predicted by $\beta c$ power in centro-frontal channels (Fig. 2c left) and by $\theta c$ power in central channels (Fig. 2e left). MTT performance gain was not predicted by $\beta c$ power (Fig. 2c right), but it was predicted by $\theta c$ power across most of the channels (Fig. 2e right).

To summarize our electrophysiological results, we also looked at the median cluster power for each electrode and evaluated how, on significant electrodes, this power related to overnight performance gain for each task. Correlations in Supplementary Table 9 suggest a direct relationship between both $\beta c$ powers in the STM night and VDT performance ($\rho(20) = 0.54$, $p = 0.017$), and $\theta c$ power in the CNT night and MTT performance ($\rho(19) = 0.57$, $p = 0.011$). To explore this further, we calculated the between-

condition (CNT-vs.-STM) differences in both spectral power and overnight performance gain and evaluated the relationships between these variables (Electrode level p-values for Power vs. Task regression in Supplementary Table 10 for $\beta c$ and Supplementary Table 11 for $\theta c$). Looking at all centro-frontal channels combined with significant power vs performance gain correlations, a linear model (i.e., %gain = $\beta_0 + \beta_1$ (cluster power (dB))) showed that CNT-vs.-STM differences in $\beta c$ power predicted between condition differences in overnight performance gain on the VDT (Model fit: $r^2 = 0.247$, $p = 0.042$, Fig. 2d). Thus, each 0.1 dB of the between condition difference in relative $\beta c$ power predicted up to an 8.7% increase in overnight performance gain

on the VDT, giving a net benefit in STM compared to CNT ($\beta_1 = 87.10$). Linear regression between condition difference in relative $\theta c$ power cluster and VDT did not reach significance ($p = 0.05$, Fig. 2f). However, an increase in CNT-vs.-STM differences in $\theta c$ power across scalp channels with significant power vs performance gain correlations predicted the increase in MTT performance gain (Model fit: $r^2 = 0.257$, $p = 0.027$, Fig. 2g). Thus, the linear model suggested around a 7% increase in overnight performance gain for each 0.1 dB of relative $\theta c$ power increase ($\beta_1 = 70.46$). These results indicate possible interactions between cortical activity and the cognitive memory processes during EM of REM sleep. Similar trends were observed when applying a stricter cluster threshold (Supplementary Fig. 7). These indicate that activations are more topographically localized, increasing associations between performance with $\theta c$ power while weakening associations between visual performance and relative $\beta c$ power.

## Discussion

This study builds directly on the early literature about auditory stimulation in REM sleep. Notably, we set out to apply stimulations on EMs; however, our post-hoc analysis shows that this was only partially successful (88.69% of our stimulations fell within REM, but only 33.24% fell on the triggering EM). We demonstrate that auditory stimulation during REM alters Rapid Eye Movements (EMs), alters neural processing and boosts some aspects of memory consolidation while impairing others across a single night of sleep. Specifically, our stimulation reduced the density of EMs while enhancing perceptual processing (measured by the VDT) and impairing procedural learning (measured by the MTT). Interestingly, the EEG spectral response was associated with the overnight consolidation of these tasks. Overall, these findings support the use of auditory stimulation to alter REM processes.

As in the earlier studies[13,15,26], auditory stimulation in REM led to a significant reduction in EM density. EM density is typically decreased following periods of REM sleep deprivation, and high REM density is thus thought to be a characteristic of relatively low sleep need[16,29]. Thus, the fact that our stimulation reduced EM density could indicate that it somehow disrupted REM sleep physiology. However, the stimulation did not wake the participants up, nor did it induce arousal. As defined by AASM, arousals during REM sleep are abrupt broadband increases in EEG activity that last for at least three seconds concurrent with submental EMG of at least one second[30]. However, our auditory stimulation did not lead to such an increase. This suggests that the auditory stimulation we used was able to modify EEG dynamics without causing a break in the continuity of the ongoing sleep period. Nevertheless, we cannot completely rule out the possibility that the stimulation caused some form of disruption which was not apparent from the usual EEG and EMG recordings. If so, this disruption was apparently favourable to the consolidation of our visual task, while at the same time, it disrupted the procedural task.

Previous studies have suggested that REM sleep duration can be increased by acoustic stimulation locked to eye movements[13–15]. However, more recent work has shown cortical responses to auditory stimuli when clicks were applied during either REM[28] or NREM sleep[31], but no alterations in sleep structure. As we found no changes in either the percentage or the total time spent in REM sleep or the time spent in phasic or tonic REM components, our results are consistent with the latter papers. These differences may be due to the small sample size used in the first studies.

Our analyses of cortical responses to auditory stimulation showed that changes in the EEG spectrum after the click predicted overnight changes in memory performance. Specifically, an increase in post-stimulus Beta power predicted improvements in VDT performance, while a decrease in post-stimulus Theta power predicted a decrease in performance on this improvement on the MTT. Main ERSP activations are also evident after applying stricter cluster thresholds. However, beta association with visual tasks seems to be marked by activations that are broader in spectrum and expanded in time, whereas theta relationships with performance were associated with more narrow band activity.

The link between theta and behavioural performance comes as no surprise. Two rodent studies which show evidence of memory reactivation in REM also found a link between reactivation and theta activity in this sleep stage[32,33]. Furthermore, optogenetic dampening of theta during REMs erased place recognition and impaired fear-conditioned contextual memory[34], while optogenetic stimulation of medial prefrontal cortex (mPFC) pyramidal neurones transiently promotes theta in association with increased EMs and phasic REM sleep[35]. In keeping with this, theta coherence between the amygdala, hippocampus, and prefrontal cortex during REM sleep predicts success in fear conditioning[36]. In humans, there is a strong relationship between theta power and wakeful memory[37,38]. Playing tones in REM sleep leads to a rapid increase in the theta band, and trials with high theta power are more strongly associated with replay[39].

Interestingly, hippocampal theta activity is known to organize place cell replay into temporal sequences and is thus important for both encoding and consolidation of spatial memories[40]. In phasic REM, when theta transiently increases, firing rates increase throughout the hippocampus and there is a greater coordination of the hippocampus and cortex, presumably allowing information exchange and consolidation[40]. Some authors have even suggested that increases in theta activity are indicative of reactivation, proposing a model whereby an increase in theta power in conjunction with a surge in spindles would facilitate consolidation[41]. Based on this literature, it seems plausible that the way in which our stimulus interfered with theta activity could have impaired memory consolidation, potentially by disrupting replay in some manner. What is less clear is why changes in theta should have opposing correlations with procedural and visual tasks; however, our own work has suggested that some reactivations may actually interfere with the consolidation of motor procedural tasks[42], so it is possible that in this case, Theta disruption would be beneficial.

It is also unclear why modulation of beta should predict consolidation on the visual task, but not the procedural task. Previous work has shown that the frontal cortices are dominated by beta activity during REM[43]. Beta bursts between the anterior cingulate and the dorsolateral prefrontal cortex, two structures associated with memory and consolidation, showed high coherence during REM. The authors speculated that beta may therefore be important for connectivity, coordinating the activity of these structures in REM. Indeed, there is evidence that visual perception learning may be facilitated by interactions of visual and prefrontal areas[44]. Thus, if beta does play a general role in connectivity, that could explain why increases in post-stimulus beta power predicted improvements in VDT. Furthermore, our Beta responses start at frequencies as low as 10 Hz which includes activations in the range of sensorimotor rhythms (SMR: 9–16 Hz)[45]. Recent work suggests that increases in SMR activity during phasic REM may be related to increased behavioural performance, mainly in central electrodes[46]. SMR is usually associated with increases in higher frequencies (Beta and Gamma)[47], so our stimulation might have enhanced common mechanisms which are beneficial for task performance. Turning to the question of localization, with our limited spatial sampling it is difficult to infer which cortical areas are involved in the

observed beta and theta dynamics. However, the topographic patterns we observe suggest that cortical activity related to changes in the MTT may spread across different regions, whereas changes in the VDT may be mediated by central and prefrontal cortices, as already suggested[48].

Several studies have evaluated the effects of REM stimulation on memory consolidation. In general, these studies use targeted memory reactivation (TMR) in which cue sounds previously associated with learned material in wake are re-presented during sleep. TMR triggers reactivation[49,50] and consolidation[51,52] when applied in NREM sleep. In a marked distinction from the TMR work, the sounds applied in our study were completely novel and unrelated to the learning tasks. Because our auditory stimulus was not associated with any memory it could not have triggered reactivation. Even so, the auditory stimulation was able to alter the REM-related memory processes, and this was likely due to the way in which the auditory stimulation impacts the pattern of neural oscillations in REM, rather than triggering memory reactivation.

We would like to recognize some limitations in our study. First, our sample size is small, and although it was enough for 80% power in the expected behavioural responses, our regression analyses are underpowered and the results are therefore not yet generalizable. Furthermore, we cannot evaluate other factors that might influence the changes in performance across conditions in the same regressions. Also, in terms of the accuracy of stimulation, our clicks occurred within REM sleep at >88% accuracy, but only occurred on the same EMs as triggered them with <50% accuracy. This suggests that our results are not necessarily linked to the EM itself, but rather to receiving the click during REM.

Commercial devices which aim to boost the cognitive function of sleep using external stimuli are now on the rise. However, we caution that these types of stimuli may decrease some of the behavioural benefits of sleep while enhancing others. Our data show this bivalent effect for stimulations during REM sleep, but further studies should address this concern and examine the potential diminishing memory-related effects of NREM sleep stimulation too.

## Methods

**Participants**. Twenty right-handed healthy participants (age: 22.75 ± 6.49, range 18–38; 7 males) with self-declared normal hearing and no history of sleep disorders completed the study. Exclusion criteria also included any use of psychoactive drugs or medications. The participants agreed not to consume caffeine or engage in extreme physical exercise within 12 h before every visit to the laboratory, and not to consume alcohol within 24 h before each visit. All participants provided written informed consent according to the ethics committee from Cardiff University School of Psychology and the Declaration of Helsinki.

**Experimental procedure**. Experimental procedures were carried out in the sleep laboratories at Cardiff University Brain Research Centre (CUBRIC). The participants spent two experimental nights in the laboratory undergoing one night with no-stimulation control condition (CNT) and one night with auditory stimulation (STM). The order of the experimental nights was balanced across subjects and separated by at least one week. Upon arrival, participants first completed a series of questionnaires including the Stanford Sleepiness Scale (SSS)[53] to determine their level of alertness and the Positive and Negative Affect Schedule (PANAS) scale[54] to evaluate their mood. All participants were trained in three different behavioural tasks: an attention task: a psychomotor vigilance test (PVT), a visual task: a visual texture discrimination test (VDT) and a procedural task: a mirror-tracing

test (MTT). These behavioural tasks were also presented in a counterbalanced order across subjects. Participants went to bed at around 11:30 p.m. After 7 h of sleep, participants had the opportunity to shower while recovering from sleep inertia. Around 1 h after awakening, performance on the behavioural tasks was tested again, in the same order as in the evening before sleep. Due to technical reasons, two participants did not complete the VDT testing in one of the four sessions. Therefore, these two subjects were excluded from the VDT analysis but were included in the electrophysiological and behavioural analyses of the other tasks. Similarly, the polysomnographic recordings of one participant were not completed during the CNT night because of the midnight failure of the EEG system. Hence, this participant was included in the comparisons of behavioural performance, but it was excluded when comparing the cortical response or sleep architecture.

**Behavioural tasks**. PVT was applied to test the sustained attention of the participants[55]. In this task, the participants were told to focus on a black background of a computer screen. A counter appeared on the centre of the screen with an inter-stimulus interval (ISI) jittered between 2 and 10 s. The participants were instructed to press the spacebar as soon as the counter appeared. Their response time was presented in milliseconds upon pressing the spacebar. This process was repeated for 10 min after which the participants had at least a 5-min break before continuing to the next task.

VDT is a texture discrimination task that involves visual learning[20]. Briefly, at the beginning of each trial, participants were asked to focus on a black screen with a fixation cross at the centre. When ready, subjects pressed the space bar after which a trial was presented. Each trial (Supplementary Fig. 1a) began with the presentation of a fixation cross at the centre of the screen (1,000 ms). Then, a stimulus image was briefly presented (17 ms). This was followed by a blank screen of varying duration (ISI of 0–400 ms) and then a masking stimulus (100 ms), composed of randomly rotated V-shaped patterns. The stimulus image consisted of a matrix of jittered lines, additionally containing two pieces of information: (i) a rotated "T" or "L" at the centre of the matrix, and (ii) an array of three diagonal lines aligned either vertically or horizontally and located in one of the four quadrants of the matrix. Then, the participants had to resolve both a letter recognition task and an orientation task. For this, using the keyboard, the subjects first indicated (i) whether the fixation point on the target screen was the letter "T" or "L" (letter recognition task), and (ii) whether the three-line array was arranged horizontally or vertically (orientation task). The position of the diagonal array was presented in different quadrants of the matrix on each experimental night to prevent learning effects from repeated testing[56]. A short high-tone sound was presented when the participants made an error in the letter task, but no feedback was presented for the orientation task.

The time interval between the target onset and the mask is referred to as the stimulus onset asynchrony (SOA). One VDT block consisted of 50 trials with a constant SOA. The task difficulty increased as the SOA decreased, and the SOA was presented in descending order. During the experiment we presented two blocks, with SOA of 400, 300, 200, and 160 ms, followed by three blocks, with SOA of 120, 100, 80, 60 and 40 ms. The outcome of each SOA block is the proportion of trials for which the letter and the orientation tasks were both correct. These values were fitted to a psychometric curve, and the outcome measure was the threshold SOA at which subjects' accuracy was 80%.

MTT was used to evaluate procedural memory[57]. Two sets of six different stimuli were used in the two experimental nights.

These stimuli were randomly assigned to each subject for test-retest on each experimental night, but the stimuli were not repeated between nights/conditions. The stimuli consisted of complex closed figures made up of 26–27 angles enclosing straight segments and one or two curved sections, such as in Supplementary Fig. 1b. The figures were centred on a black background covering $600 \times 600$ pixels with 27 pixel-wide traces. The figures were presented using a custom-made MATLAB script that included instructions for the participant. Following the mechanism of the mirror-tracing apparatus, the movement of the cursor was mirrored vertically: downward movements of the mouse translated to upward movements of the cursor and vice versa. Using the non-dominant left hand, the participants were asked to move the computer mouse to follow the trace of the figure shape without moving out of the path. The participants were able to see the trace of the path when moving the mouse. Different trace colours were presented when the cursor was on-the-path or out-of-the-path. An error consisted of moving the cursor off the path of a figure and the script counted error time once the cursor was beyond the thick line of the figure. In each trial, the error time was not measured separately for single errors but was rather accumulated to the total error time[58].

In the learning condition before sleep, a star-shaped figure was used for the initial training. The participant traced the star, starting and ending at the same marked point. The mirror-tracing of the star was repeated until the subject reached a criterion of no more than ten errors and a minimum of 80% of the pixels of the mouse trace kept on-path. Then, the six line-drawn experimental figures were presented one after the other. In the recall condition in the morning after sleep, the star was presented first to warm up the subjects and to keep the conditions comparable, but the star figure was tested only once. Then, the six line-drawn experimental figures were presented in a randomized order. The subject traced each figure starting at the top and ending at the top. Performance was evaluated as the time of the cursor on the path[57].

**Polysomnographic recordings**. Standard polysomnography consisting of EEG, chin EMG and EOG were continuously recorded using passive Ag/AgCl electrodes and collected with a BrainAmp DC amplifier (Brain Products). EEG electrodes were positioned at 21 scalp sites according to the international 10–20 system (Fpz, Fp1, Fp2, Fz, F3, F4, F7, F8, Cz, C3, C4, Pz, P3, P4, P7, P8, Oz, O1, O2, T7, and T8) and referenced to Cpz. Impedances were maintained below 5 kΩ. The data were sampled at 500 Hz and saved without further filtering. An offline re-referencing to the mastoids (M1, M2) was then applied.

Sleep stages were scored manually for each 30-s epoch according to the ASSM criteria[30]. Sleep scoring was performed by two trained experimenters blinded to stimulation periods and using a publicly available interface (https://github.com/mnavarretem/psgScore). All artefacts and arousals were visually identified and marked in the hypnogram. The total sleep duration was defined as the time between the first transition from wake to any sleep stage until the last transition from any sleep stage to the last wake after sleep onset (WASO). Total sleep time is the total time in any sleep stage other than wake. Sleep efficiency was defined as the percentage of total sleep time from the total sleep duration[59]. Finally, phasic and tonic REM sleep were defined based on REM epochs from the hypnogram with (phasic REM) or without (tonic REM) eye saccades[60].

**Acoustic stimulation and eye movement detection**. We implemented a semi-automatic algorithm for auditory stimulation applied locked to the EM in the EOG during REM. Stimulation timestamps were recorded online when the clicks were applied. For the STM condition, the acoustic stimuli consisted of stereophonic clicks of pink noise (50 ms duration) with rising and falling slopes (5 ms duration each). In the CNT condition, the detection protocol was identical to the STM condition, but the clicks were muted. For the online detection algorithm, the left EOG signal was filtered using a Chebyshev type II passband filter (3 dB at $f = 0.3$ Hz and $f = 5$ Hz; >50 dB at $f < 0.1$ Hz and $f > 15$ Hz). Each acoustical stimulus was applied 100 ms after the filtered EOG signal crossed a detection threshold indicating an EM. This detection threshold was tuned by the experimenter depending on the voltage amplitudes of the ongoing REM saccades during the first REM period. Thus, the threshold was set for the absolute amplitude of the saccades between the 50–100 μV interval and was not further modified during the rest of the night. Selection of the EOG detection channel was done in either the left or right electrode depending on signal-to-noise ratio and saccade amplitudes. The saccade detection was paused for 2 s after each stimulus. The algorithm was turned on by the experimenter when at least two or three EM and reduced chin activity appeared during sleep. The stimulation was likewise turned off by the experimenter when any signs of arousal or sleep stage changes were evident in the online recordings.

**Data pre-processing**. EEG recordings were analysed with MATLAB using the Fieldtrip toolbox[61], together with custom-made functions. EEG channels were filtered between 0.3–35 Hz using a zero-phase Chebyshev Type II bandpass filter (3 dB at 0.16 and 35.8 Hz; >100 dB at $f < 0.05$ Hz and $f > 45$ Hz). Continuous data was segmented into trials encompassing the period from 2 s pre-stimulus to 3 s post-stimulus. Stimuli applied during any sleep stage other than REM were excluded from the analysis (e.g., clicks applied during epochs scored as Wake). Likewise, any trial that overlapped with or was closer than 2 s to arousal or an artefact was removed ($n < 5\%$ of all trials). Trials with noisy channels were corrected using spherical interpolation, but only if the noisy channels accounted for less than 25% of the total scalp electrodes. Trials with noisy channels accounting for more than 25% of the total scalp electrodes were removed. Supplementary Table 1 presents a summary of applied and selected trials. Then, the EEG data was denoised using the Extended Infomax ICA algorithm[62,63]. Briefly, with minimal preprocessing, we visually removed segments with infrequent atypical artefacts. Next, we performed ICA decomposition using Fieldtrip's extended 'runica' method for each recording and used topographic distribution to identify the components that reflect eye and heart artefacts. Finally, we removed noise components and back-projected to the data. EOG and EMG channels were used offline to check whether the auditory stimulation was applied correctly, i.e., locked to REM eye movements, and then compared between conditions of stimulation to account for possible EOG or EMG disruptions. Thus, EOG channels were filtered between 0.5–8 Hz using a zero-phase Chebyshev Type II bandpass filter (3 dB at 0.36 and 8.2 Hz; >100 dB at $f < 0.26$ Hz and $f > 10$ Hz). Similarly, EMG channels were filtered between 10–100 Hz using a zero-phase Chebyshev Type II bandpass filter (3 dB at 9.56 and 100.35 Hz; >100 dB at $f < 8.97$ Hz and $f > 105$ Hz) (Supplementary Figs. 2 and 5).

The spectral response of each trial was computed using a complex Morlet Wavelet of 6 cycles for frequencies between 0.25 and 30 Hz in steps of 0.25 Hz. After computing the time-frequency components for each trial, only the sections ranging from 1 s pre-stimulus to 2.5 s post-stimulus were kept for further analysis to avoid boundary effects. Finally, a single-trial baseline correction for event-related spectral decomposition was applied using a baseline interval of 1 s before the stimulus onset. We used decibels as a

comparison of power levels. Then, the time–frequency power was normalized using a decibel transform (dB) (dB power $= 10$ log10 [power/baseline])[64].

**Behavioural data analysis**. Independent performance measures were computed for each task, as described below. For the PVT task, the task performance was determined as the mean response time (RT) for valid trials. Valid trials were defined as responses with RT between 100 and 500 ms[55]. For the VDT task, the performance was defined as the threshold SOA at which subjects' accuracy was equal to 80%. Briefly, the percentage of correct responses for the orientation task was calculated for each SOA in the test session. Then, a cumulative Gaussian function was fitted to obtain a psychometric curve and thus determine the threshold SOA that corresponded to the 80% correct performance. Psychometric functions were fitted using psignifit (v.2.5.6) in MATLAB (see http://bootstrap-software.org/psignifit/) which implements the maximum-likelihood method[65]. All trials in which the letter task was incorrect were removed from the threshold SOA calculation[7]. For the MTT task, the performance was considered as the logarithm of the time when the cursor was on-path (performance log-time, LT). We removed trials with log-time values larger than 3.5 SD away from the group mean, which represented outliers or failed attempts to draw the figure. A total of two trials were thus removed from all trials/subjects. Finally, for all tasks, the overnight performance gain was computed as the relative change of performance between sessions (Overnight performance gain $= 100 \times$ (evening performance–morning performance)/(evening performance)).

**Statistics and reproducibility**. All figures show a mean $\pm$ 95% confidence interval (CI) unless stated otherwise. All pairwise comparisons for sleep architecture and changes in performance were computed using paired tests unless said otherwise. Paired-sample $t$-test was conducted for normally distributed data, as indicated by the Shapiro–Francia test[66]. Odd ratios were converted using a logit transformation before the $t$-test to compare between paired samples. Wilcoxon signed-rank test was used for comparisons presenting non-Gaussian distribution.

Multivariate linear regression analyses were used as explanatory models to determine the unique contributions of the sleep macro-architecture and conditions of stimulation (STM and CNT) to changes in memory performance. These multivariable regressions were defined as Performance gain (%) $= \beta_0 + \beta_1$(Condition) $+ \beta_2$(Sleep Stage) and allowed us to determine the independent contribution of both explanatory variables to the changes in behavioural performance[67].

Significant differences between conditions of stimulation for event-related potentials (ERPs) and induced event-related spectral perturbation (ERSP) responses were computed using nonparametric suprathreshold cluster test[68,69]. For this, we first averaged the ERP and ERSP responses of all EEG electrodes for each subject and then compared them between conditions. Clusters were determined by adjacent ERP or ERSP values with $t$-values representing a $p$-value $< 0.05$, computed by a two-tailed Welch's $t$-test. Then, the permutation distribution of the maximal suprathreshold cluster size measured in pixels was calculated by re-labelling values of CNT and STM conditions using 1600 non-repeated permutations[70]. The significance of the cluster size was evaluated by comparing the suprathreshold cluster size to the permutation distribution[68]. Between-condition differences for CNT-vs-STM were obtained by subtracting the measured variable (time or performance) on the night of the STM condition from the night of the CNT condition (CNT–STM). Positive values therefore indicate larger

magnitudes for STM, and negative values indicate larger magnitudes for CNT. Between-condition differences for frequency power were calculated as the ratio of power during the STM night and the CNT night, followed by a log transformation to decibels (dB power $= 10$ $\log_{10}$ [$power_{STM}/power_{CNT}$]). Values $> 0$ indicate larger magnitudes for STM, and values lower than 0 indicate larger magnitudes for CNT.

Subsequently, we evaluated the topographic incidence of the detected clusters within the EEG electrodes that may be associated with the performance gain on behavioural tasks. The average power of each identified cluster was computed for each electrode and associated with the performance gain on the behavioural tasks. These electrode-wise associations were evaluated using linear regressions defined as Performance gain (%) $= \beta_0 + \beta_1$ [cluster Power (dB)]. Only electrodes with significant linear fits and surviving correction for multiple comparisons were selected as channels associated with the behavioural performance. To summarize the joint effect of all significant electrodes, we used linear regressions to estimate how changes in the average cluster power across significant channels predicted the changes in performance gain.

We computed the false discovery rate (FDR) to control for multiple comparisons using the Benjamini–Hochberg method at $q < 0.05$[71]. To reduce the effect of outliers present in the data, all linear regressions were fitted with robust linear models using Huber's M-estimator[72]. All statistical analyses were computed using MATLAB R2018b.

Lastly, we performed an a priori power analysis using G*Power version 3.1.9.7 to determine the minimum sample size required to test matched differences between CNT and STM without replication. We focused on the estimation of the sample size based on the behavioural responses. This analysis indicated that the sample size required to achieve 80% power detecting a two-tailed Cohen's $d$ effect size of 0.7, at a significance criterion of $\alpha = 0.05$, is $N = 19$. Considering this initial sample size for our behavioural analyses, an estimation for Multiple Linear Regression with two predictors indicated $N = 20$ as the sample size necessary to achieve a 60% power and $\rho^2$ of 0.3 at a significance criterion of $\alpha = 0.05$.

**Reporting summary**. Further information on research design is available in the Nature Portfolio Reporting Summary linked to this article.

## Data availability

The numerical source data behind the graphs in Figs. 1c–g, 2d, f, g can be found in Supplementary Data 1. The anonymised experimental data that support the findings of this study are available from OSF repository[73] https://osf.io/bjeq2/ with the https://doi.org/10.17605/OSF.IO/BJEQ2.

## Code availability

The custom MATLAB code is provided as an additional folder available in the OSF repository[73] https://osf.io/bjeq2/.

Published online: 16 Februrary 2024

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

## Acknowledgements
The authors would like to thank Lorena Santamaría, Sofia Pereira, Niall McGinley and Ibad Kashif for their help with participants' recruitment and sleep scoring. This work was funded by the ERC grant SolutionSleep, 681607, to P.L. We also thank all the participants for their time and commitment to the study.

## Author contributions
M.N.: Conceptualization, methodology, software, validation, formal analysis, investigation, resources, data curation, writing—original draft, writing—review & editing, visualization. VG: Validation, formal analysis, investigation, resources, data curation, writing —original draft, writing—review & editing. M.R.: Validation, formal analysis, investigation, resources, data curation, writing—original draft, writing—review & editing. M.B.: Conceptualization, methodology, investigation, writing—review & editing, supervision. P.L.: Conceptualization, methodology, resources, writing—review & editing, supervision, project administration, funding acquisition.

## Competing interests
The authors declare no competing interests.
