## [Peer Review File · Communications Biology]

Reviewers' comments:

Reviewer #1 (Remarks to the Author):

General comment:

In this study, Navarrete and colleagues sought to influence mechanisms of memory consolidation by intervening on REM sleep. This is an innovative approach as most studies investigating the mechanisms of sleep-mediated memory consolidation have focused on NREM sleep rather than REM sleep. In practice, Navarrete and colleagues asked healthy participants to perform two different cognitive tasks (visual and procedural respectively) before and after sleep. During sleep, acoustic stimuli were played following Rapid Eye-Movements (EMs) (STM condition) or no sounds were played (CNT condition). The authors report a modification of brain activity, recorded with EEG, in the STM condition compared to the CNT condition with an increase in beta power and a decrease in theta power. They also report an improvement in the visual task but a deterioration of performance in the procedural task in the STM conditions whereas no effect of memory consolidation was observed in the CNT condition. While the overall approach is interesting, the motivation of this study appears a bit unclear: why were these two tasks selected? Why targeting EMs? It is also unclear if the authors successfully modulated REM sleep or just disrupted it with their intervention. Finally, a mechanistic explanation for the pattern of results obtained is somewhat missing.

Main Comments:

1. The motivations for the choice of methodology do not appear clearly. First, the authors tested two different memory tasks. Why focusing on a visual and procedural task? Were the authors expecting different outcomes for these tasks and, if so, what were their hypotheses? Are these tasks known to benefit from a sleep-mediated memory consolidation? If so, why did the authors fail to observe a gain in performance in the CNT condition? This absence of memory consolidation in the CNT condition is very problematic for a study seeking to modulate memory consolidation. Second, the authors decided to time-lock acoustic stimuli to EMs. Why were EMs targeted? My understanding is that the authors hoped to target PGO waves through EMs but the literature on PGO waves suggest that EMs are a consequence of PGO waves so it is unclear how targeting EMs would affect PGO waves. Besides, the link between PGO waves and memory consolidation is not very well supported.

2. I have some concerns regarding the intervention used in this study. Indeed, the authors mention in their introduction that, contrary to NREM sleep, there are few (if any) options to modulate REM sleep. However, it is unclear if their approach does modulate REM sleep or merely perturbs it. Indeed, by playing acoustic stimuli following EMs they observed fewer EMs, an increase in faster frequencies and a decrease in theta rhythms, a pattern of results which could be interpreted as inducing a form of arousals. The authors did check that the acoustic stimulations did not induce changes on EOG and EMG recordings, but the EMG data was analyzed only on a 2.5s window following stimulations (Fig. S5a). Figure S5b stops at 30Hz which is too low in frequency to observed modulations of muscle tone. And Figure S6b only examines the proportion of epochs with very high EMG activity ($>\text{mean} + 3.5\text{SD}$). Even if no peripheral arousal was observed (in the EOG and EMG), the EEG shows a response compatible with an arousal. If the intervention just perturbed REM sleep, then the interpretation of the results could significantly change as the effects on memory could reflect a consequence of arousal rather than a modulation of REM-mediated processes. The absence of memory consolidation in the CNT condition also suggests that the memory effects might not be REM-dependent. Finally, I am wondering why the authors choose a no-stimulation condition as a control. They could also have played sounds but not time-locked to EMs to show the specificity of the time-locking to EMs in terms of EEG changes or memory performance. By having no stimulation in the control condition, it is unclear if the results are linked to the EM-locking or any stimulations in REM sleep would have the same impact.

3. I have reservations regarding some of the statistical analyses and results presented in this study. The strategy to correct for multiple comparisons is unclear. First, in Table 1, were tests corrected for multiple comparisons? Second, in Figure 2, the authors report multiple dependent tests (correlation between beta and theta power and performance gain in the visual and procedural tasks). While the tests by electrodes seem to have been FDR-corrected, were the tests in Figure 2d,f,g also corrected? The test reported in 2f does not cross the significance threshold and the test in 2d is very close to that threshold. Given that these correlations are obtained on electrodes passing a FDR-corrected threshold (from what I understood) the low correlation values and high p-values are surprising. The left topography of Figure 2e also shows many significant electrodes (after a FDR correction) but low r^2 values (around 0.2 considering the color bar, which would correspond to uncorrected p-values around 0.025). For the avoidance of doubt, would it be possible to provide the uncorrected p-values for these topographies (Figure 2c and 2e) as well as the value of the corrected thresholds?

4. Each participants participated twice for the STM and CNT conditions. Since the authors expected a long-term memory effect of task practice, did they check if there was a modification of the performance between the first and second session (above and beyond the STM and CNT conditions)? Should this be included in statistical analyses?

Minor Comments:

1. "REM sleep is critical for memory, emotion, and cognition". The absence of consequence of REM suppression (e.g., through brain lesions or pharmacological treatments) make it hard to consider REM sleep "critical" for these cognitive functions.
2. "the huge PGO waves which have been strongly associated with memory consolidations". It could be argued that PGO waves are not huge, since they have not been observed in non-invasive recordings in humans! Also, the reference 22 does not appear to mention PGO waves and, to my knowledge, PGO waves have not been "strongly" associated with memory consolidation.
3. I am surprised the authors do not report an effect of stimulations on the P2 amplitude since the stimuli triggered a P2 that was absent otherwise. Figure 2a also shows a rather clear P2 effect. Perhaps, the FDR correction used is not optimal here.
4. "this frequency [theta] dominates REM". This might be true in rodents but not in humans since there are not theta peaks in scalp EEG during REM sleep.
5. The authors rely on the literature on hippocampal theta to interpret these results, but the hippocampal theta recorded invasively in rodents and the cortical theta recorded non-invasively in humans are very likely completely different neural phenomena.
6. How exactly were data cleaned using ICA?
7. Were the power computations made "a priori" that is before testing? If so, why did the authors choose a low statistical power of 60% for the MLR analysis?
8. I have a few minor questions about the EM detection algorithm:
 - a. Auditory stimulations were delivered according to the amplitude of eye movement calibrated during the first cycle and REM period. How was the first "REM period" identified? How long was it? How many EMs were detected in this period?
 - b. How did the authors ensure that stimulations were delivered only in REM sleep and not in other sleep stages?
 - c. Were both left and right EMs used (so a negative and positive threshold)? Or just one side?
 - d. Did the authors check that there was no stimulation outside REM sleep or that the algorithms did not confound EMs and sleep slow waves?

Reviewer #2 (Remarks to the Author):

In a sample of 20 young healthy participants, Navarrete et al. compared found differential effects of auditory stimulation timed to eye movements during REM (compared with no stimulation) on learning/memory consolidation in different domains, whereby auditory stimulation improved perceptual processing (as measured by a visual texture discrimination task) and impaired procedural learning (as measured by a mirror tracing task). Visual task performance was associated with higher beta power in ERSPs from auditory stimuli, whereas motor task performance was associated with higher theta power. Auditory stimulation also lowered eye movement density.

I enjoyed reading this manuscript and learning about these novel results. My one and perhaps only major concern is that because the stimulation was only compared to a control condition in which the bursts of auditory pink noise were muted, the authors cannot infer whether timing the bursts to eye movements was important ... what would have happened if the control group had instead received randomly timed stimulation? Below are some thoughts that I'd like the authors to address, falling under the above concern:

- 1) The justification for timing stimuli to eye movements during REM is that the eye movements should correspond to POG waves. Of the two references supporting this claim (21 and 22), I did not find a mention of POG waves in the second reference (Frazer et al. 2021). Could the authors add more references supporting this which clearly discuss POG waves?
- 2) The authors used EOG and EMG channels to check offline whether auditory stimuli were accurately timed to eye movements. As far as I can tell though, the accuracy of the algorithm used to time these stimuli to eye movements is never reported. I think it would be useful to report the accuracy as a percentage of trials that were correctly timed, as well as the false positive and false negative rate. If reporting this for all data is too labor intensive, the authors could use n minutes of data randomly sampled from REM for each subject, where n is a manageable data length to estimate the accuracy (e.g., 5 minutes).
- 3) Going along with the prior comment, I would like to see whether the accuracy in each individual subject can be used to predict learning effects within the experimental condition. It would be interesting to know if higher accuracy of the stimulus timing predicts a greater impact on learning; here, even a trend would be useful to illuminate whether the timing of the stimuli is really important or not.
- 4) Also building on comment #2, how often were auditory clicks triggered during NREM sleep?
- 5) The authors write that the EOG channels were filtered 0.3 – 5 Hz in real time to detect eye movements. I wonder if 50 dB of stopband attenuation at 15 Hz is too aggressive? This would seem to imply that the rising phase of the saccade should last \gg 67 ms, which seems too long given the boxcar-like waveform of saccades. Did the authors experiment with different filtering parameters during a pilot phase of the experiment to determine the parameters used here, in particular, the lowpass frequency cutoff?

Besides these comments, I have a few minor points which I'd like to ask the authors to address.

- 1) The noisy channel threshold for rejecting trials was 25%. This threshold actually seems quite high ... how did the authors arrive at this value?
- 2) The authors used an IIR filter (Chebyshev Type II) for offline processing and analysis of EEG signals. IIR makes sense for real time applications (e.g., triggering stimuli), but it seems like an odd choice for offline applications. Why did the authors decide to use this filter and not an FIR filter with better performance?
- 3) As far as I can tell, the authors didn't covary for ordering effect in the models. If this is added to the models (e.g., grouping based on those who received stimulation in the first versus second session), does it improve model fit (e.g., as judged by a log-likelihood ratio test)?

- 4) The authors used a cluster threshold of t-stats corresponding to $P < 0.05$. How sensitive are the results to this threshold? It's well known that the choice of threshold is fairly arbitrary for cluster permutation stats and, in principle, any threshold can be used as long as it's also applied to the permuted data. Some have argued that a critical test statistic value (e.g., corresponding to $P = 0.05$) should not be used as a threshold, since the cluster stats approach shouldn't be rooted in parametric statistics. I would be curious what the results look like for much higher or much lower thresholds (or, optionally, using threshold-free cluster enhancement, Smith and Nichols 2009 Neuroimage).
- 5) Although the authors did a statistical power analysis to arrive at the sample size of $N = 20$, I think it should be acknowledged that this is a rather modest sample size, particularly for correlating EEG power with behavior.
- 6) Building on the above comment, the age is reported as 22.76 ± 6.94 years. Is this \pm STD? I assume that age was not normally distributed, otherwise something like one quarter of the participants were younger than 18 years. Could you please report the age range (min to max)? Regardless, this is a rather young sample (average age of 22 years), which should probably also be stated clearly up front.
- 7) The participants were asked to refrain from caffeine or alcohol before each session. Were they also instructed or asked about use of other substances that affect REM, such as cannabis? Were any participants taking sleep medications?

Dear Reviewers: Many thanks for these useful comments. We include a point-by-point response below.

Reviewer #1 (Remarks to the Author):

General comment:

In this study, Navarrete and colleagues sought to influence mechanisms of memory consolidation by intervening on REM sleep. This is an innovative approach as most studies investigating the mechanisms of sleep-mediated memory consolidation have focused on NREM sleep rather than REM sleep. In practice, Navarrete and colleagues asked healthy participants to perform two different cognitive tasks (visual and procedural respectively) before and after sleep. During sleep, acoustic stimuli were played following Rapid Eye-Movements (EMs) (STM condition) or no sounds were played (CNT condition). The authors report a modification of brain activity, recorded with EEG, in the STM condition compared to the CNT condition with an increase in beta power and a decrease in theta power. They also report an improvement in the visual task but a deterioration of performance in the procedural task in the STM conditions whereas no effect of memory consolidation was observed in the CNT condition.

While the overall approach is interesting, the motivation of this study appears a bit unclear: why were these two tasks selected?

Our tasks were chosen because previous studies have demonstrated that REM sleep plays a role in their consolidation, and we thus predicted that manipulations of REM would also impact on this. Thus, the VDT has been shown to be sensitive to REM (Stickgold et al., 2000; Tamaki et al., 2020) and reactivate during REM (Yotsumoto et al., 2009). The MTT is also associated with REM e.g. (Plihal and Born, 1997), and learning this task leads to increased REM density (Fogel et al., 2007). An interesting model proposes that early consolidation of the MTT depends on REM sleep, while later consolidation (once the skill is already quite well learned) depends on NREM sleep (Smith et al., 2004).

To clarify this matter, we included in the text:

‘In the current study we build on this literature by exploring the ways in which EM-triggered auditory stimulation actively alters REM electrophysiology, and how such stimulation impacts on the consolidation of non-auditory REM dependent memory tasks for which replay could not be triggered by the click stimuli. We predict that such auditory stimulation would lead to changes in EEG activity, which would in turn predict cognitive performance. To test this, we examined memory performance and the associated post-stimulus spectral EEG activity in two tasks that have been previously associated with REM related consolidation: the visual texture discrimination task (VDT)^{7,19,20}, and the mirror tracing task (MTT)⁸. Recent studies have suggested that REM sleep helps to stabilize previous learning of the VDT, possibly by regulating synaptic activity during REM^{7,21}. Regarding the MTT, learning of this task has been shown to increase REM density²², and an interesting model proposes that early consolidation of the MTT depends on REM sleep, while later consolidation (once the skill is already quite well learned) depends on NREM sleep²³. Others have shown that high acetylcholine levels and associated increases in REM density during REM, triggered by an acetylcholinesterase inhibitor, facilitate enhanced sleep-dependent MTT gains^{24,25}’.

Why targeting EMs?

We chose to use EMs to trigger our auditory stimulation because we were building upon an older literature in which a myriad of experiments has used EMs to trigger auditory stimulation during REM periods. These previous works (Guerrien et al., 1989; Mouze-Amady et al., 1986; Sockeel et al., 1987) suggested that auditory stimulation during EMs could modulate EM density and suggested that this could lead to changes in memory processing (Guerrien et al., 1989).

Importantly, a number of studies also managed to manipulate REM through auditory stimulation that was not locked to EMS, e.g. (Arankowsky-Sandoval et al., 1992; Salin-Pascual et al., 1994; Vazquez et al., 1998). In most cases this was successful and increased REM duration while reducing EMS. However a direct comparison of the EM linked and non-EM linked method showed that the former was significantly more effective when it came to boosting memory consolidation for a morse code comparison task (Guerrien et al., 1989). To our knowledge this was the only study which looked at the impact of such stimulation on memory consolidation. The result was very encouraging, however the authors cautioned that because their REM stimulation used the same modality as their morse code learning task it might have reinstated the task. Thus, our current experimentation, examining two REM-linked tasks, and characterising the physiological response to stimulation, was necessary.

To clarify this matter, we included the following in our introduction:

‘A series of studies from the 80s and 90s provided initial evidence that it is possible to alter electrophysiological characteristics of REM through sensory stimulation. These early studies suggested that brief auditory stimulation triggered by REM Eye Movements (EMs) can increase the length of REM sleep in humans¹³⁻¹⁵, while also reducing the density of EMs^{13,14}. Importantly, a number of studies also managed to manipulate REM through auditory stimulation that was not locked to EMS¹⁶⁻¹⁸. In most cases this increased REM duration while reducing EMs. However, a direct comparison of the EM linked and non-EM linked method showed that the former was significantly more effective when it came to boosting memory consolidation for a morse code comparison task¹⁵. To our knowledge this was the only study which looked at the impact of such stimulation on memory consolidation. Although the result was very encouraging, the authors cautioned that because their REM stimulation used the same auditory modality as their morse code learning task it might have actually reinstated the task.

In the current study we build on this literature by exploring the ways in which EM-triggered auditory stimulation actively alters REM electrophysiology, and how such stimulation impacts on the consolidation of non-auditory REM dependent memory tasks for which replay could not be triggered by the click stimuli. We predict that such auditory stimulation would lead to changes in EEG activity, which would in turn predict cognitive performance. To test this, we examined memory performance and the associated post-stimulus spectral EEG activity in two tasks that have been previously associated with REM related consolidation: the visual texture discrimination task (VDT)^{7,19,20}, and the mirror tracing task (MTT)⁸. Recent studies have suggested that REM sleep helps to stabilize previous learning of the VDT, possibly by regulating synaptic activity during REM^{7,21}. Regarding the MTT, learning of this task has been shown to increase REM density²², and an interesting model proposes that early consolidation of the MTT depends on REM sleep, while later consolidation (once the skill is already quite well learned) depends on NREM sleep²³. Others have shown that high acetylcholine levels and associated increases in REM density during REM, triggered by an acetylcholinesterase inhibitor, facilitate enhanced sleep-dependent MTT gains^{24,25}.

Given the prior literature, we hypothesized that auditory stimulation triggered by EMs would lead to changes in REM length, number of EMs, and EEG activity, which would in turn alter consolidation of our REM sensitive memory tasks. Our findings revealed that auditory stimulation can indeed modulate both EEG and behavioural consolidation. Interestingly, our data also suggest that this stimulation method might enhance some memory processes while disturbing others.’

It is also unclear if the authors successfully modulated REM sleep or just disrupted it with their intervention.

Sensory stimulation can of course disrupt sleep continuity. However, in response to this concern, we conducted a series of additional analyses, and found no evidence for the stimulation causing arousals in our examinations of TST, NREM, and REM, EEG responses, or EMG. Please see our response to Main Point #2 (from this reviewer, see below). Below for details on these analyses.

We note, however, that while consolidation of the MTT was disrupted by this manipulation, consolidation of the VDT was facilitated. Thus, if the manipulation *did* cause a disruption then it is clear that the disruption either occurred *in addition to* other processes which were beneficial for the VDT or was itself somehow beneficial.

We have added the below text to this effect:

‘As defined by AASM, arousals during REM sleep are abrupt broadband increases in EEG activity that last for at least three seconds concurrent with submental EMG of at least one second³⁰. However, our auditory stimulation did not lead to such an increase. This suggests that the auditory stimulation we used was able to modify EEG dynamics without causing a break in the continuity of the ongoing sleep period. Nevertheless, we cannot completely rule out the possibility that the stimulation caused some form of disruption which was not apparent from the usual EEG and EMG recordings. If so, this disruption was apparently favourable to the consolidation of our visual task, while at the same time, it disrupted the procedural task.’

Finally, a mechanistic explanation for the pattern of results obtained is somewhat missing.

Main Comments:

1. The motivations for the choice of methodology do not appear clearly. First, the authors tested two different memory tasks. Why focusing on a visual and procedural task? Were the authors expecting different outcomes for these tasks and, if so, what were their hypotheses? Are these tasks known to benefit from a sleep-mediated memory consolidation? If so, why did the authors fail to observe a gain in performance in the CNT condition? This absence of memory consolidation in the CNT condition is very problematic for a study seeking to modulate memory consolidation.

We chose to use EMs to trigger our auditory stimulation because we were building upon an older literature in which a myriad of experiments has used EMs to trigger auditory stimulation during REM periods. These previous works (Guerrien et al., 1989; Mouze-Amady et al., 1986; Sockeel et al., 1987) suggested that auditory stimulation during EMs could modulate EM density and suggested that this could lead to changes in memory processing (Guerrien et al., 1989). We have elaborated further on this in our response (2) above. In that response, we also cite the amendments made to the introduction to set this out clearly in the manuscript.

Notably, we did not expect different results from these two tasks. Instead, we expected to enhance both by increasing REM duration. As it was, our results are more complex (and more interesting) than expected, since they suggest that different subprocesses within REM are important for consolidation of Procedural and Visual tasks.

Although the MTT and VDT are strongly linked to REM, this stage does not always produce a notable benefit if it is not manipulated, see (Rasch and Born, 2013) for a review. Instead, the manipulation of REM has been shown to impact on these tasks, which is why we chose them.

Second, the authors decided to time-lock acoustic stimuli to EMs. Why were EMs targeted? My understanding is that the authors hoped to target PGO waves through EMs but the literature on PGO waves suggest that EMs are a consequence of PGO waves so it is unclear how targeting EMs would affect PGO waves. Besides, the link between PGO waves and memory consolidation is not very well supported.

We apologise if our original text erroneously gave the impression that we were aiming to target PGO waves. This was not our intention since, instead, our use of EMs as a trigger for the stimulation merely followed the literature using this same technique. We have now clarified this in the introduction as cited above (2nd query from this reviewer).

2. I have some concerns regarding the intervention used in this study. Indeed, the authors mention in their introduction that, contrary to NREM sleep, there are few (if any) options to modulate REM sleep. However, it is unclear if their approach does modulate REM sleep or merely perturbs it. Indeed, by playing acoustic stimuli following EMs they observed fewer EMs, an increase in faster frequencies and a decrease in theta rhythms, a pattern of results which could be interpreted as inducing a form of arousals. The authors did check that the acoustic stimulations did not induce changes on EOG and EMG recordings, but the EMG data was analyzed only on a 2.5s window following stimulations (Fig. S5a). Figure S5b stops at 30Hz which is too low in frequency to observed modulations of muscle tone. And Figure S6b only examines the proportion of epochs with very high EMG activity (>mean + 3.5SD). Even if no peripheral arousal was observed (in the EOG and EMG), the EEG shows a response compatible with an arousal. If the intervention just perturbed REM sleep, then the interpretation of the results could significantly change as the effects on memory could reflect a consequence of arousal rather than a modulation of REM-mediated processes.

Thanks for explaining this concern. In response, we have further analyses to check for the possibility that our manipulation perturbs sleep.

To this end, we first assessed wake time, arousals, and microarousals for TST, NREM, and REM (separately), finding no evidence for sleep disruption in any of these (see below tables):

Uncorrected p values for TST events.

	CNT mean	CNT_95CI	STM mean	STM 95CI	pVal	tVal	df
Wake	9.2	3.1	7.7	3.1	0.4	0.9	19.0
Arousal	126.8	25.2	132.8	17.8	0.7	-0.4	19.0
Microarousal	9.2	25.2	7.7	17.8	0.9	-0.1	19.0

Uncorrected p values for NREM events

	CNT mean	CNT_95CI	STM mean	STM 95CI	pVal	tVal	df
Wake	4.9	2.2	3.8	1.8	0.4	0.8	19
Arousal	109.1	24.4	121.5	17.3	0.3	-1.0	19
Microarousal	4.9	24.4	3.8	17.3	0.6	-0.5	19

Uncorrected p values for REM events

	CNT mean	CNT_95CI	STM mean	STM 95CI	pVal	tVal	df
Wake	0.85	0.56	0.25	0.24	0.07	1.93	19
Arousal	3.75	4.27	1.1	2.78	0.18	1.38	19
Microarousal	0.85	4.27	0.25	2.78	0.23	1.24	19

We next performed a new time-frequency analysis on our EEG data using a bigger window (-1.5 to 4 sec post stimulus and 0 to 100 Hz). The below topoplots show no significant differences between Stim and Sham in any analysis. The stats for each electrode are in the tables underneath.

T values (CNT - STM)

Uncorrected p values for TST events

Channel	Events during total sleep time (p values)			Events during NREM (p values)			Events during REM (p values)		
	Wake	Arousal	Micro-arousal	Wake	Arousal	Micro-arousal	Wake	Arousal	Micro-arousal
Fp1	0.178	0.923	0.197	0.106	0.458	0.078	0.246	0.087	0.376
Fpz	0.235	0.903	0.297	0.343	0.420	0.129	0.285	0.203	0.222
Fp2	0.325	0.864	0.752	0.273	0.642	0.313	0.248	0.163	0.073
F7	0.882	0.895	0.413	0.956	0.672	0.211	0.243	0.036	0.179
F3	0.619	0.318	0.149	0.753	0.076	0.047	0.297	0.081	0.043
Fz	0.323	0.285	0.075	0.513	0.077	0.014	0.428	0.082	0.066
F4	0.424	0.227	0.297	0.913	0.056	0.105	0.529	0.148	0.111
F8	0.421	0.628	0.202	0.443	0.325	0.055	0.131	0.280	0.172
T7	0.507	0.759	0.509	0.147	0.460	0.193	0.555	0.180	0.093
C3	0.182	0.526	0.325	0.441	0.146	0.118	0.044	0.125	0.122
Cz	0.162	0.187	0.231	0.705	0.015	0.061	0.011	0.086	0.091
C4	0.480	0.614	0.398	0.600	0.135	0.144	0.083	0.100	0.044
T8	0.604	0.623	0.841	0.394	0.965	0.765	0.197	0.176	0.215
P7	0.343	0.788	0.415	0.353	0.816	0.180	0.065	0.153	0.289
P3	0.486	0.742	0.441	0.539	0.967	0.209	0.040	0.285	0.499
Pz	0.359	0.839	0.425	0.427	0.673	0.253	0.049	0.242	0.379
P4	0.352	0.911	0.730	0.199	0.565	0.305	0.226	0.169	0.450

P8	0.618	0.716	0.422	0.268	0.499	0.188	0.551	0.091	0.389
O1	0.610	0.896	0.892	0.718	0.985	0.818	0.867	0.594	0.678
Oz	0.293	0.921	0.393	0.305	0.605	0.240	0.285	0.659	0.621
O2	0.375	0.874	0.443	0.238	0.825	0.251	0.871	0.452	0.467

Notably, while the tables do show significant results at some electrodes, none of these would survive correction for multiple comparisons. And as we have no reason to focus on a specific channel, we cannot consider results at individual electrodes.

Next, following the reviewers' concern, we performed a new time-frequency analysis on our EMG data using a bigger window (-1.5 to 4 sec post stimulus and 0 to 100 Hz).

EMG TF for all stimulations during REM sleep RESULTS:

To summarise the results shown in the above figures – while there is a non-significant EMG response to stimulation (A), this does not survive cluster correction (C). Thus, there is no evidence of greater arousal or disruption to sleep after our sound cues were played.

In summary, we conducted a series of additional analyses to test for the possibility that our stimulation led to arousals. These found no evidence for such arousals in our examinations of TST, NREM, and REM, EEG responses, or EMG. Also, we demonstrated that the stimulation did not cause sleep disruption as arousals defined by the AASM rules which are based on clinical consensus. Despite these negative results, we acknowledge that some form of disruption may have occurred even though it is not apparent in any of our analyses. We note, however, that while consolidation of the MTT was disrupted by this manipulation, consolidation of the VDT was facilitated. Thus, if the manipulation *did* cause a disruption then it is clear that the disruption either occurred *in addition to* other processes which were beneficial for the VDT or was itself somehow beneficial. We have added the below text to this effect:

‘As defined by AASM, arousals during REM sleep are abrupt broadband increases in EEG activity that last for at least three seconds concurrent with submental EMG of at least one second³⁰. However, our auditory stimulation did not lead to such an increase. This suggests that the auditory stimulation we used was able to modify EEG dynamics without causing a break in the continuity of the ongoing sleep period. Nevertheless, we cannot completely rule out the possibility that the stimulation caused some form of disruption which was not apparent from the usual EEG and EMG recordings. If so, this disruption was apparently favourable to the consolidation of our visual task, while at the same time, it disrupted the procedural task.’

3. The absence of memory consolidation in the CNT condition also suggests that the memory effects might not be REM-dependent.

While the MTT and VDT are certainly sensitive to REM (see (Stickgold et al., 2000; Tamaki et al., 2020; Yotsumoto et al., 2009) for the VDT, and (Plihal and Born, 1997), and (Fogel et al., 2007) for the MTT), the picture is far from simple. The MTT does not always consolidate across undisturbed REM (for example in (Hornung et al., 2007; Rasch et al., 2009) while artificially enhancing REM (in these cases via pharmacology) has been linked to improvements on this task (Hornung et al., 2007), but see also results from Rasch et al 2009 where pharmacological suppression of REM lead to an improvement on this task. The situation is similar for the VDT, since the control group (with unmanipulated sleep) frequency shows no evidence of consolidation (Yotsumoto et al., 2009). In fact, recent work has argued for complementary roles of NREM and REM sleep, with the former providing a highly plastic milieu conducive to general performance enhancement and the latter offering stabilization of these offline gains by rendering them resistant to retrograde interference (Tamaki et al., 2020). Our current observation that manipulating REM led to a benefit for one task (the VDT) and a deficit in the other (MTT) provides further evidence for a complex relationship between REM and the consolidation of these tasks. Thus, we would not necessarily expect an improvement in performance on these tasks after a normal period of REM, instead we hypothesised that manipulating REM would lead to this type of improvement.

4. Finally, I am wondering why the authors choose a no-stimulation condition as a control. They could also have played sounds but not time-locked to EMs to show the specificity of the time-locking to EMs in terms of EEG changes or memory performance. By having no stimulation in the control condition, it is unclear if the results are linked to the EM-locking or any stimulations in REM sleep would have the same impact.

This is an important point, and we apologise that our initial write-up of the manuscript did not make the answer clear. In this paper, we set out to build on existing work that stimulated EMs in REM using auditory clicks. We chose we chose to use clicks that were timed to EMs, e.g. (Arankowsky-Sandoval et al., 1992; Salin-Pascual et al., 1994; Vazquez et al., 1998), because a direct comparison of the EM linked and non-EM linked method showed that the former was significantly more effective when it came to boosting memory consolidation for a morse code comparison task (Guerrien et al., 1989). While we agree that it might also have been interesting to compare stimulation locked to EMs to non EM locked stimulation, this was not the goal of our study. Instead, we simply wished to characterise

the impacts of EM locked stimulation on behavioural consolidation and EEG. Thus, comparison with non-EM-locked stimulation was outside the scope of our study.

5. I have reservations regarding some of the statistical analyses and results presented in this study. The strategy to correct for multiple comparisons is unclear. First, in Table 1, were tests corrected for multiple comparisons? Second, in Figure 2, the authors report multiple dependent tests (correlation between beta and theta power and performance gain in the visual and procedural tasks). While the tests by electrodes seem to have been FDR-corrected, were the tests in Figure 2d,f,g also corrected? The test reported in 2f does not cross the significance threshold and the test in 2d is very close to that threshold. Given that these correlations are obtained on electrodes passing a FDR-corrected threshold (from what I understood) the low correlation values and high p-values are surprising. The left topography of Figure 2e also shows many significant electrodes (after a FDR correction) but low r^2 values (around 0.2 considering the colour bar, which would correspond to uncorrected p-values around 0.025). For the avoidance of doubt, would it be possible to provide the uncorrected p-values for these topographies (Figure 2c and 2e) as well as the value of the corrected thresholds?

For table 1: the statistics were not corrected for multiple comparisons as this is a main descriptive table, now we clarified that these are uncorrected values.

For figures 2: d,f and g, these were not corrected across them because the hypothesis are different in each of the statistic. However, p values for fig 2d and 2g survive an FDR correction when correcting for multiple comparisons across these regressions (Q threshold = 0.05; m = 3):

p val	rank	(i/m)*Q	var <= q?
0.027	1	0.0167	FALSE
0.042	2	0.0333	FALSE
0.05	3	0.0500	TRUE

P critical after FDR correction is therefore $q = 0.05$.

Now, as the reviewer suggest, we present both uncorrected and corrected p values of the channel regression for each of the cases:

For Figure 2c

Uncorrected p values									
Beta - Visual Task					Beta - Procedural Task				
	0.0046	0.0034	0.0020		0.1303	0.0279	0.1036		
0.0035	0.0080	0.0014	0.0015	0.0153	0.0293	0.2787	0.1403	0.0965	0.1861
0.1146	0.0196	0.0010	0.0012	0.0012	0.1611	0.1115	0.0953	0.0323	0.1951
0.0703	0.0854	0.0581	0.0126	0.0132	0.4729	0.3334	0.1572	0.1195	0.2300
	0.0832	0.2825	0.0658		0.5728	0.5537	0.0553		
Corrected p values									
Beta - Visual Task					Beta - Procedural Task				
	0.0108	0.0091	0.0070		0.2602	0.2259	0.2602		
0.0091	0.0167	0.0062	0.0062	0.0247	0.2259	0.3443	0.2602	0.2602	0.2732
0.1203	0.0294	0.0062	0.0062	0.0062	0.2602	0.2602	0.2602	0.2259	0.2732
0.0869	0.0944	0.0813	0.0230	0.0230	0.5226	0.3890	0.2602	0.2602	0.3018
	0.0944	0.2825	0.0864		0.5728	0.5728	0.2602		

Channel reference									
	Fp1	Fpz	Fp2			Fp1	Fpz	Fp2	
F7	F3	Fz	F4	F8	F7	F3	Fz	F4	F8
T7	C3	Cz	C4	T8	T7	C3	Cz	C4	T8
P7	P3	Pz	P4	P8	P7	P3	Pz	P4	P8
	O1	Oz	O2			O1	Oz	O2	

For figure 2e

Uncorrected p values									
Beta - Visual Task					Beta - Procedural Task				
	0.0421	0.0409	0.0498			0.0334	0.0060	0.0179	
0.0019	0.0109	0.0285	0.0188	0.0149	0.0391	0.0229	0.0188	0.0138	0.0255
0.0233	0.0015	0.0048	0.0019	0.0085	0.0039	0.0252	0.0033	0.0309	0.0501
0.2791	0.0014	0.0055	0.0012	0.1930	0.0075	0.0191	0.0817	0.0278	0.3804
	0.7228	0.3388	0.0930			0.0110	0.0007	0.0021	
Corrected p values									
Beta - Visual Task					Beta - Procedural Task				
	0.0590	0.0590	0.0654			0.0412	0.0253	0.0364	
0.0080	0.0254	0.0461	0.0359	0.0312	0.0456	0.0382	0.0364	0.0362	0.0382
0.0407	0.0080	0.0166	0.0080	0.0222	0.0204	0.0382	0.0204	0.0406	0.0553
0.3084	0.0080	0.0166	0.0080	0.2251	0.0264	0.0364	0.0858	0.0390	0.3804
	0.7228	0.3557	0.1149			0.0329	0.0148	0.0204	
Channel reference									
	Fp1	Fpz	Fp2			Fp1	Fpz	Fp2	
F7	F3	Fz	F4	F8	F7	F3	Fz	F4	F8
T7	C3	Cz	C4	T8	T7	C3	Cz	C4	T8
P7	P3	Pz	P4	P8	P7	P3	Pz	P4	P8
	O1	Oz	O2			O1	Oz	O2	

We added this information as supplementary tables S8 and S9 and included in the text:

‘To explore this further, we calculated the between-condition (CNT-vs-STM) differences in both spectral power and overnight performance gain and evaluated the relationships between these variables (Electrode level p-values for Power vs Task regression in Table S8 for beta and Table S9 for theta).’

6. Each participants participated twice for the STM and CNT conditions. Since the authors expected a long-term memory effect of task practice, did they check if there was a modification of the performance between the first and second session (above and beyond the STM and CNT conditions)? Should this be included in statistical analyses?

Thanks for this question. We did examine session as a factor, but this was not significant. Please see below for our ANOVAs examining the statistics for Session Time / condition:

For VDT when computing in terms of percentage gain
ANOVA Table (type II tests)

Effect	DFn	DFd	F	p	p<.05	ges
Condition	1	32	12.446	0.001	*	0.280000
Session	1	32	0.005	0.942		0.000167
condition:session	1	32	2.306	0.139		0.067000

For MTT when computing in terms of percentage gain
ANOVA Table (type II tests)

Effect	DFn	DFd	F	p	p<.05	ges
Condition	1	36	5.183	0.029	*	0.126
Session	1	36	1.722	0.198		0.046
condition: Session	1	36	1.083	0.305		0.029

We included this information in the text:

‘Additionally, a two-way ANOVA was used to determine the effect of Condition (CNT vs STM) and Learning Order (Night 1 vs Night 2) on the performance of each task. For the VDT this ANOVA revealed that there was no significant interaction between the effects of Condition and Learning Order ($F(1, 32) = 2.31, p = 0.139$), and ~~only~~ the only Simple Main Effect was that of Condition on VDT performance (Condition $p = .001$, Learning Order $p = 0.942$). A similar two-way ANOVA for MTT showed no interactions between the effects of Condition and Learning Order ($F(1, 36) = 1.08, p = 0.305$), and only Condition presented a significant Simple Main Effect on MTT performance (Condition $p = .029$, Learning Order $p = 0.305$).’

Minor Comments:

1. “REM sleep is critical for memory, emotion, and cognition”. The absence of consequence of REM suppression (e.g., through brain lesions or pharmacological treatments) make it hard to consider REM sleep “critical” for these cognitive functions.

We have rephrased this as ‘REM sleep is believed to be important for memory, emotion, and cognition’.

2. “the huge PGO waves which have been strongly associated with memory consolidations”. It could be argued that PGO waves are not huge, since they have not been observed in non-invasive recordings in humans! Also, the reference 22 does not appear to mention PGO waves and, to my knowledge, PGO waves have not been “strongly” associated with memory consolidation.

We have removed this sentence.

3. I am surprised the authors do not report an effect of stimulations on the P2 amplitude since the stimuli triggered a P2 that was absent otherwise. Figure 2a also shows a rather clear P2 effect. Perhaps, the FDR correction used is not optimal here.

As the reviewer points out, we did find a response to stimulation which is shown in Figure 2. However, we must recognize that the linked mastoid reference used for analysis may also not be the most appropriate technique in order to do a proper ERP analysis compared to others such as single electrode

or average references (Clayson et al., 2021; Hu et al., 2019). However, our ERP analysis is showed here as a reference for the reader but not as a central point of our aims, so we preferred to maintain the linked mastoid reference to keep the channel balance within the global activity of sleep. Therefore, we suspect that the failure of this response to reach significance probably just a power/noise issue.

4. “this frequency [theta] dominates REM”. This might be true in rodents but not in humans since there are not theta peaks in scalp EEG during REM sleep.

Thanks for flagging this. We have removed this sentence, please see the point below for the new section inserted here.

5. The authors rely on the literature on hippocampal theta to interpret these results, but the hippocampal theta recorded invasively in rodents and the cortical theta recorded non-invasively in humans are very likely completely different neural phenomena.

We have extended our discussion of Theta to include more of the recent work on humans. Please see below:

‘Furthermore, optogenetic dampening of theta during REMs erased place recognition and impaired fear-conditioned contextual memory³⁴, while optogenetic stimulation of medial prefrontal cortex (mPFC) pyramidal neurones transiently promoted theta in association with increased EMs and phasic REM sleep³⁵. In keeping with this, theta coherence between the amygdala, hippocampus, and prefrontal cortex during REM sleep predicts success in fear conditioning³⁶. In humans, there is a strong relationship between theta power and wakeful memory^{37,38}. Playing tones in REM sleep leads to a rapid increase in theta band, and trials with high theta power are more strongly associated with replay³⁹.’

6. How exactly were data cleaned using ICA?

We used the same procedure for Cleaning artifacts using ICA as proposed by the developers of the fieldtrip toolbox (https://www.fieldtriptoolbox.org/tutorial/ica_artifact_cleaning/). Therefore, for each independent night:

1. Read the data with minimal preprocessing using `ft_preprocessing`
2. Remove segments with infrequent atypical artifacts using either `ft_rejectvisual`. In this section, we paid main attention to data variance.
3. ICA decomposition of the data using `ft_componentanalysis` using method: ‘runica’. An additional step included computing the correlation of the ICA component and the eye movement on the EOG. High correlation of the components and the EOG suggested eye artifacts in the EEG.
4. Identifying the components that reflect eye and heart artifacts using `ft_topoplotIC`.
5. Removing those components and back projecting the data using `ft_rejectcomponent`.

We included a summary in the text:

‘Then, the EEG data was denoised using the Extended Infomax ICA algorithm^{62,63}. Briefly, with minimal preprocessing, we visually removed segments with infrequent atypical artefacts. Next, we performed ICA decomposition using Fieldtrip’s extended ‘runica’ method for each recording, and used topographic distribution to identify the components that reflect eye and heart artefacts. Finally, we removed noise components and back projected the data. EOG and EMG channels were used offline to check whether the auditory stimulation was applied correctly, i.e., locked to REM eye movements, and then compared between conditions of stimulation to account for possible EOG or EMG disruptions.’

7. Were the power computations made “a priori” that is before testing? If so, why did the authors choose a low statistical power of 60% for the MLR analysis?

We chose our sample size with the behavioural results in mind. Previous studies off closed loop stimulation during sleep range in effect size from 0.3 – 1.0 [see (Harlow et al., 2023) for a recent review] and we therefore considered our number good enough for evidencing behavioural changes. This analysis indicated that the sample size required to achieve 80% power detecting a two-tailed Cohen’s d effect size of 0.7, at a significance criterion of $\alpha = .05$, is $N = 19$. We recognize that our study was a bit underpowered for regression analysis; however, we wanted to be transparent about this fact and thus we also reported the conditions for these regressions. Consequently, we changed the wording of the claim and recognize this situation in our methods as we also recognize the limitations of these analyses:

‘Lastly, we performed an a priori power analysis using G*Power version 3.1.9.7 to determine the minimum sample size required to test matched differences between CNT and STM. We focused the estimation of the sample size based on the behavioural responses. This analysis indicated that the sample size required to achieve 80% power detecting a two-tailed Cohen’s d effect size of 0.7, at a significance criterion of $\alpha = .05$, is $N = 19$. Considering this initial sample size for our behavioural analyses, an estimation for Multiple Linear Regression with two predictors indicated $N = 20$ as the samples size necessary to achieve a 60% power and p^2 of 0.3 at a significance criterion of $\alpha = .05$.’

Also, we included a new section recognizing this limitation:

‘We would like to recognize some limitations in our study. First, our sample size is small, and although it was enough for 80% power in the expected behavioural responses, our regression analyses are underpowered, and the results are therefore not yet generalizable. Furthermore, we cannot evaluate other factors that might influence the changes in performance across conditions in the same regressions.’

8. I have a few minor questions about the EM detection algorithm:

a. Auditory stimulations were delivered according to the amplitude of eye movement calibrated during the first cycle and REM period. How was the first “REM period” identified? How long was it? How many EMs were detected in this period?

REM cycles were identified visually, when first two to three rapid eye movements were evident. We included this in the text:

‘The algorithm was turned on by the experimenter when at least two or three EM and reduced chin activity appeared during sleep. The stimulation was likewise turned off by the experimenter when any signs of arousal or sleep stage changes was evident in the online recordings.’

b. How did the authors ensure that stimulations were delivered only in REM sleep and not in other sleep stages?

Automatic stimulation was turned on visually when rapid eye movements were evident for the experimenter. At any evidence of arousal or change of sleep stage (e.g., sleep spindles, slow eye movements, blinking, etc) the automatic stimulation was turned off. We have further clarified this in the method’s text:

‘We implemented a semi-automatic algorithm for auditory stimulation applied locked to the EM in the EOG during REM. Stimulation timestamps were recorded online when the clicks were applied. For the STM condition, the acoustic stimuli consisted of stereophonic clicks of

pink noise (50 ms duration) with rising and falling slopes (5 ms duration each). In the CNT condition, the detection protocol was identical to the STM condition, but the clicks were muted. For the online detection algorithm, the left EOG signal was filtered using a Chebyshev type II passband filter (3 dB at $f = 0.3$ Hz and $f = 5$ Hz; >50 dB at $f < 0.1$ Hz and $f > 15$ Hz). Each acoustical stimulus was applied 100 ms after the filtered EOG signal crossed a detection threshold indicating an EM. This detection threshold was tuned by the experimenter depending on the voltage amplitudes of the ongoing REM saccades during the first REM period. Thus, the threshold was set for the absolute amplitude of the saccades between the 50-100 μ V interval and was not further modified during the rest of the night. Selection of the EOG detection channel was done in either left or right electrode depending on signal to noise ratio and saccade amplitudes. The saccade detection was paused for 2 s after each stimulus. The algorithm was turned on by the experimenter when at least two or three EM and reduced chin activity appeared during sleep. The stimulation was likewise turned off by the experimenter when any signs of arousal or sleep stage changes were evident in the online recordings.'

c. Were both left and right EMs used (so a negative and positive threshold)? Or just one side?

Yes, left, and right EMGs were used depending in which had better signal or amplitude quality. Now we clarified this in the text:

'We implemented a semi-automatic algorithm for auditory stimulation applied locked to the EM in the EOG during REM. Stimulation timestamps were recorded online when the clicks were applied. For the STM condition, the acoustic stimuli consisted of stereophonic clicks of pink noise (50 ms duration) with rising and falling slopes (5 ms duration each). In the CNT condition, the detection protocol was identical to the STM condition, but the clicks were muted. For the online detection algorithm, the left EOG signal was filtered using a Chebyshev type II passband filter (3 dB at $f = 0.3$ Hz and $f = 5$ Hz; >50 dB at $f < 0.1$ Hz and $f > 15$ Hz). Each acoustical stimulus was applied 100 ms after the filtered EOG signal crossed a detection threshold indicating an EM. This detection threshold was tuned by the experimenter depending on the voltage amplitudes of the ongoing REM saccades during the first REM period. Thus, the threshold was set for the absolute amplitude of the saccades between the 50-100 μ V interval and was not further modified during the rest of the night. Selection of the EOG detection channel was done in either left or right electrode depending on signal to noise ratio and saccade amplitudes. The saccade detection was paused for 2 s after each stimulus. The algorithm was turned on by the experimenter when at least two or three EM and reduced chin activity appeared during sleep. The stimulation was likewise turned off by the experimenter when any signs of arousal or sleep stage changes were evident in the online recordings.'

d. Did the authors check that there was no stimulation outside REM sleep or that the algorithms did not confound EMs and sleep slow waves?

The algorithm was turned off in any presence of spindles or Slow Waves. We have checked the stimulation accuracy and reported on table below:

	mean	std	median	iqr
ACC for clicks during REM sleep	88.69	12.86	92.52	10.61
ACC for EM events during total sleep time (Including REM, NREM, and wake)	30.07	14.53	32.22	21.63
ACC for EM events during REM	33.24	15.66	36.71	24.62

This indicates that in average 88.69% of the clicks fall in stages of REM sleep. We would like to note that while the accuracy for stimulations on EM in REM is only 33%, our method was similar to that of the prior literature which we are following (Guerrien et al., 1989; Mouze-Amady et al., 1986; Sockeel

et al., 1987). These papers don't report the accuracy, but we have no reason to expect it would have been any higher than ours. No clicks were played during Slow waves and clicks in other sleep stages mainly corresponded to epochs in stage transitions.

We have specified this in the text as follows:

‘Our stimulation setup was manually turned-on in the presence of EMs during REM sleep and turned-off in the presence of any arousal or changes in the sleep stage. During the CNT night the same protocol was run but sounds were muted (Figure 1b and Figure S2). This resulted in an average of $88.69\% \pm 1.86\%$ of clicks applied during REM sleep, and $33.24\% \pm 15.66\%$ of clicks applied on the EM that triggered the stimulus. ‘

Reviewer #2 (Remarks to the Author):

In a sample of 20 young healthy participants, Navarrete et al. compared found differential effects of auditory stimulation timed to eye movements during REM (compared with no stimulation) on learning/memory consolidation in different domains, whereby auditory stimulation improved perceptual processing (as measured by a visual texture discrimination task) and impaired procedural learning (as measured by a mirror tracing task). Visual task performance was associated with higher beta power in ERSPs from auditory stimuli, whereas motor task performance was associated with higher theta power. Auditory stimulation also lowered eye movement density.

I enjoyed reading this manuscript and learning about these novel results. My one and perhaps only major concern is that because the stimulation was only compared to a control condition in which the bursts of auditory pink noise were muted, the authors cannot infer whether timing the bursts to eye movements was important ... what would have happened if the control group had instead received randomly timed stimulation? Below are some thoughts that I'd like the authors to address, falling under the above concern:

This is an important point, and we apologise that our initial write-up of the manuscript did not make the answer clear. In this paper, we set out to build on existing work that stimulated EMs in REM using auditory clicks. We chose we chose to use clicks that were timed to EMs, e.g. (Arankowsky-Sandoval et al., 1992; Salin-Pascual et al., 1994; Vazquez et al., 1998), because a direct comparison of the EM linked and non-EM linked method showed that the former was significantly more effective when it came to boosting memory consolidation for a morse code comparison task (Guerrien et al., 1989). While we agree that it might also have been interesting to compare stimulation locked to EMs to non-EM locked stimulation, this was not the goal of our study. Instead, we simply wished to characterise the impacts of EM locked stimulation on behavioural consolidation and EEG. Thus, comparison with non-EM-locked stimulation was outside the scope of our study.

1) The justification for timing stimuli to eye movements during REM is that the eye movements should correspond to POG waves. Of the two references supporting this claim (21 and 22), I did not find a mention of POG waves in the second reference (Frazer et al. 2021). Could the authors add more references supporting this which clearly discuss POG waves?

We locked our stimulation to EMs because we were following the prior literature which had done this with great success. We have removed our discussion of PGO waves, as we agreed that it is difficult to know the exact relationship between these and EMs.

2) The authors used EOG and EMG channels to check offline whether auditory stimuli were accurately timed to eye movements. As far as I can tell though, the accuracy of the algorithm used to time these stimuli to eye movements is never reported. I think it would be useful to report the accuracy as a percentage of trials that were correctly timed, as well as the false positive and false negative rate. If

reporting this for all data is too labour intensive, the authors could use n minutes of data randomly sampled from REM for each subject, where n is a manageable data length to estimate the accuracy (e.g., 5 minutes).

We apologise for this oversight and have now calculated the accuracy for REM/TST/and EM in REM (please see the table below)

Clicks accuracy (%).

	mean	std	median	iqr
ACC for clicks during REM sleep	88.69	12.86	92.52	10.61
ACC for EM events during total sleep time (Including REM, NREM, and wake)	30.07	14.53	32.22	21.63
ACC for EM events during REM	33.24	15.66	36.71	24.62

We would like to note that while the accuracy for stimulations on EM in REM is only 33%, we used a method very similar to the papers (Guerrien et al., 1989; Mouze-Amady et al., 1986; Sockeel et al., 1987) in the prior literature which we are following. These papers don't report the accuracy, but we have no reason to expect it would have been any higher than ours.

3) Going along with the prior comment, I would like to see whether the accuracy in each individual subject can be used to predict learning effects within the experimental condition. It would be interesting to know if higher accuracy of the stimulus timing predicts a greater impact on learning; here, even a trend would be useful to illuminate whether the timing of the stimuli is really important or not.

Thanks for this suggestion. We have run this analysis and summarise the results below. Interestingly, for the MTT, the % EMs stimulated predicted a decrease in performance change across the experimental night for the MTT. However, this performance change was not correlated with the total number of clicks applied during REM sleep. This could suggest that disruption of the procedural task is associated with induced activity during the EMs. Other task changes (e.g. VDT and PVT) were not correlated with either total clicks in REM or total EM stimulated:

rho Coeff	% EMs stimulated during REM sleep	% clicks stimulated on EMs during total sleep time	% clicks stimulated on EMs during REM sleep time
VDT change	-0.27	0.00	0.10
MTT change	-0.43	-0.57	-0.54
PVT change	0.25	-0.09	-0.13

P values	% EMs stimulated during REM sleep	% clicks stimulated on EMs during total sleep time	% clicks stimulated on EMs during REM sleep time
VDT change	0.282	0.999	0.695
MTT change	0.067	0.011	0.016
PVT change	0.311	0.720	0.594

We have added this information to the text as follows:

'The percentage of stimulated EMs was correlated with the overnight change in performance for MTT (Pearson's $\rho = -0.54$, $p_{uncorr} = .016$) but not for VDT (Pearson's $\rho = 0.10$, $p_{uncorr} =$

.695). However, MTT performance was not correlated with the total number of clicks applied during REM sleep (Pearson's $\rho = -0.43$, $p_{uncorr} = .067$).

4) Also building on comment #2, how often were auditory clicks triggered during NREM sleep?

In average, 20.45% of stimulation clicks were applied in NREM stages (N1) scored offline (median 18.67%, range 9.31% to 38.83%). Although these were mostly applied during transitions and changes of sleep stage. For these, clicks, they had an average delay between them of 17.84s (median 14.94s)

5) The authors write that the EOG channels were filtered 0.3 – 5 Hz in real time to detect eye movements. I wonder if 50 dB of stopband attenuation at 15 Hz is too aggressive? This would seem to imply that the rising phase of the saccade should last $\gg 67$ ms, which seems too long given the boxcar-like waveform of saccades. Did the authors experiment with different filtering parameters during a pilot phase of the experiment to determine the parameters used here, in particular, the lowpass frequency cutoff?

We chose these parameters based on (Agarwal et al., 2005) which used them to detect REM offline. It is important to highlight that the bandpass was defined at 3dB, which means that activities on cutoff frequencies are half-amplitude from the original. Briefly, anything beyond these cut-offs might be considered reduced enough to be comparable to the original activity. However, these band-rejected activities can still be present as an unwanted noise to the focus frequencies in the bandpass that might affect the signal detection whatever the attenuation band of the filter. Therefore, having a stringent filter can help to reduce this unwanted noise. We recognize that stringent attenuation bands may affect considerably the bandpass frequencies if the filters are not designed carefully or if using other filter types which are typical in EEG analysis (e.g., Butterworth or elliptic). For this reason, we used a zero-phase Chebyshev Type II bandpass filter with second order sections which is a stable filter with the ability to control the attenuation ripple in the bandpass frequencies. The lowpass cutoff was furthermore evaluated on simulations and considered good enough to the experimenter to recognize the saccades and the algorithm to detect the event. However, we acknowledge that not many studies have detected REM online, and there may be room for improvement.

‘... Also, in terms of the accuracy of stimulation, our clicks occurred within REM sleep at $>88\%$ accuracy, but only occurred on the same EMs that they were triggered by with $<50\%$ accuracy. This suggests that our results are not necessarily linked to the EM itself, but rather to receiving the click during REM.’

Besides these comments, I have a few minor points which I'd like to ask the authors to address.

1) The noisy channel threshold for rejecting trials was 25%. This threshold actually seems quite high ... how did the authors arrive at this value?

We would like to emphasise that we used overnight recordings of >8 hours. Participants were free to move around in their beds, and in this situation, it is not unusual to have epochs of noise. As we were interested in the spectral response of stimulation, removing a trial if more than a quarter of channels were noisy was a way to keep quality of the signals and reducing spurious noise activity. However, we were able to keep 77% and 73% of trials in stim and sham respectively, and we feel this is quite reasonable. Therefore, we think that the set threshold gave us a good trade-off between signal quality and trial quantity.

2) The authors used an IIR filter (Chebyshev Type II) for offline processing and analysis of EEG signals. IIR makes sense for real time applications (e.g., triggering stimuli), but it seems like an odd choice for offline applications. Why did the authors decide to use this filter and not an FIR filter with better performance?

Filter design characteristics were defined considering the narrow band characteristics of sleep EEG and filter stability. It is true that FIR filters have been proposed to be optimal filters for single trial analysis with particular interest in ERP components (Widmann et al., 2014). FIR advantages over IIR filters are mostly associated with the linear phase and they're always stable regardless of the filter order. However, FIR filter requires always extremely high orders to equal the transition band of IIR filter (e.g., 253th - order in (Warby et al., 2014)) as well as higher computing resources under the same bandwidth conditions. Therefore, optimal performance would be more related to the application rather than the type of filter recursion. As FIR filters have a linear phase, they are optimal to applications where phase prediction is required to be precise across all the frequency band including bandwidth cutoffs (e.g., (Navarrete et al., 2020)). However, in our current analysis linear phase analysis is not required as we were more interested in central frequency activities.

We implemented a Chebyshev type II filter using second order section which has multiple advantages over other IIR and FIR designs. First, the second-order sections filter design provides efficient stability (Barnes, 1985). Second, the Chebyshev type II filter design provides an efficient way to control the transition band as well as a maximally flat frequency response in the passband, which makes it the closest to an ideal filter, and minimal Gibbs phenomena (Herrmann, 1971). Third, as a recursive filter with a second-order section design, this filter was computationally more efficient than any FIR filter. This memory efficiency is important considering the size of each channel recording is on average larger than 6 hours and channels were filtered across all the recording to minimize border effects for the trials. Lastly, this filter type has been previously used in the analysis of sleep patterns (Riedner et al., 2007), epilepsy (Valderrama et al., 2012), and speech perception (Porbadnigk et al., 2013) among others.

3) As far as I can tell, the authors didn't covary for ordering effect in the models. If this is added to the models (e.g., grouping based on those who received stimulation in the first versus second session), does it improve model fit (e.g., as judged by a log-likelihood ratio test)?

We thank the reviewer for this suggestion. Unfortunately, if we were to use 'order' as a covariate it would decrease our power markedly. However, we have looked at this question in a different way by examining 'session effects' and found nothing significant (see tables below). While session training might influence performance (Julius and Adi-Japha, 2016; Snoddy, 1926), this did not influence our results of interest. We avoided session training effect by counterbalancing the CNT and STM nights as well by changing figure complexity in the MTT and the focus quadrant for the VDT (Karni and Sagi, 1991).

For VDT when computing in terms of percentage gain

ANOVA Table (type II tests)

Effect	DFn	DFd	F	p	p<.05	ges
Condition	1	32	12.446	0.001	*	0.280000
Session	1	32	0.005	0.942		0.000167
condition:session	1	32	2.306	0.139		0.067000

For MTT when computing in terms of percentage gain

ANOVA Table (type II tests)

Effect	DFn	DFd	F	p	p<.05	ges
Condition	1	36	5.183	0.029	*	0.126
Session	1	36	1.722	0.198		0.046
condition: Session	1	36	1.083	0.305		0.029

We have added this to the text as follows:

‘Additionally, a two-way ANOVA was used to determine the effect of Condition (CNT vs STM) and Learning Order (Night 1 vs Night 2) on the performance of each task. For the VDT this ANOVA revealed that there was no significant interaction between the effects of Condition and Learning Order ($F(1, 32) = 2.31, p = 0.139$), and the only Simple Main Effect was that of Condition on VDT performance (Condition $p = .001$, Learning Order $p = 0.942$). A similar two-way ANOVA for MTT showed no interactions between the effects of Condition and Learning Order ($F(1, 36) = 1.08, p = 0.305$), and only Condition presented a significant Simple Main Effect on MTT performance (Condition $p = .029$, Learning Order $p = 0.305$).’

4) The authors used a cluster threshold of t-stats corresponding to $P < 0.05$. How sensitive are the results to this threshold? It’s well known that the choice of threshold is fairly arbitrary for cluster permutation stats and, in principle, any threshold can be used as long as it’s also applied to the permuted data. Some have argued that a critical test statistic value (e.g., corresponding to $P = 0.05$) should not be used as a threshold, since the cluster stats approach shouldn’t be rooted in parametric statistics. I would be curious what the results look like for much higher or much lower thresholds (or, optionally, using threshold-free cluster enhancement, Smith and Nichols 2009 Neuroimage).

We used $p < 0.05$ because it is arguably the most standard threshold for this kind of analysis. We acknowledge that other thresholds can sometimes be appropriate, but we have no a priori hypothesis about why a different threshold would be better here.

5) Although the authors did a statistical power analysis to arrive at the sample size of $N = 20$, I think it should be acknowledged that this is a rather modest sample size, particularly for correlating EEG power with behaviour.

The reviewer is correct, and we also recognize that this is a modest sample size. Therefore, now we recognize this more clearly in our limitations section as follows:

‘We would like to recognize some limitations in our study. First, our sample size is small, and although it was enough for 80% power in the expected behavioural responses, our regression analyses are underpowered and the results are therefore not yet generalizable. Furthermore, we cannot evaluate other factors that might influence the changes in performance across conditions in the same regressions. In terms of the accuracy of stimulation, our clicks occurred within REM sleep at $>88\%$ accuracy, but only occurred on the same EMs as triggered them with $<50\%$ accuracy. This suggests that our results are not necessarily linked to the EM itself, but rather to receiving the click during REM.’

6) Building on the above comment, the age is reported as 22.76 ± 6.94 years. Is this \pm STD? I assume that age was not normally distributed, otherwise something like one quarter of the participants were younger than 18 years. Could you please report the age range (min to max)? Regardless, this is a rather young sample (average age of 22 years), which should probably also be stated clearly up front.

Yes, the reviewer is correct as the age distribution is not normal and this is mean \pm std. We thank the reviewer for pointing out this information as we realized there is a typo, and this should correspond to 22.75 ± 6.49 years. The median is 20 years and range [18 – 38 years]. This is now corrected and more specified in the text:

‘Twenty right-handed healthy participants (age: 22.75 ± 6.49 , range 18 – 38; 7 males) with self-declared normal hearing and no history of sleep disorders completed the study.’

7) The participants were asked to refrain from caffeine or alcohol before each session. Were they also instructed or asked about use of other substances that affect REM, such as cannabis? Were any participants taking sleep medications?

Yes – we used these as exclusion criteria. Thus, participants using psychoactive drugs or medications were not accepted into the study:

‘... Exclusion criteria also included any use of psychoactive drugs or medications. The participants agreed not to consume caffeine or engage in extreme physical exercise within 12 hours before every visit to the laboratory, and not to consume alcohol within 24 hours before each visit.’

REFERENCES

- Agarwal R, Takeuchi T, Laroche S, Gotman J (2005) Detection of rapid-eye movements in sleep studies. *IEEE Trans Biomed Eng* 52:1390–1396.
- Arankowsky-Sandoval G, Stone WS, Gold PE (1992) Enhancement of REM sleep with auditory stimulation in young and old rats. *Brain Res* 589:353–357.
- Barnes C (1985) A parametric approach to the realization of second-order digital filter sections. *IEEE Trans Circuits Syst* 32:530–539.
- Clayson PE, Baldwin SA, Rocha HA, Larson MJ (2021) The data-processing multiverse of event-related potentials (ERPs): A roadmap for the optimization and standardization of ERP processing and reduction pipelines. *Neuroimage* 245:118712.
- Fogel SM, Smith CT, Cote KA (2007) Dissociable learning-dependent changes in REM and non-REM sleep in declarative and procedural memory systems. *Behav Brain Res* 180:48–61.
- Guerrien A, Dujardin K, Mandal O, Sockeel P, Leconte P (1989) Enhancement of memory by auditory stimulation during postlearning REM sleep in humans. *Physiol Behav* 45:947–950.
- Harlow TJ, Jané MB, Read HL, Chrobak JJ (2023) Memory retention following acoustic stimulation in slow-wave sleep: a meta-analytic review of replicability and measurement quality. *Front Sleep* 2.
- Herrmann O (1971) On the approximation problem in nonrecursive digital filter design. *Circuit Theory, IEEE Trans* 18:411–413.
- Hornung OP, Regen F, Danker-Hopfe H, Schredl M, Heuser I (2007) The Relationship Between REM Sleep and Memory Consolidation in Old Age and Effects of Cholinergic Medication. *Biol Psychiatry* 61:750–757.
- Hu S, Yao D, Bringas-Vega ML, Qin Y, Valdes-Sosa PA (2019) The Statistics of EEG Unipolar References: Derivations and Properties. *Brain Topogr* 32:696–703.
- Iber C, Ancoli-Israel S, Chesson Jr. AL, Quan SFS, Chesson A, Quan SFS (2007) *The AASM Manual for the Scoring of Sleep and Associated Events: Rules Terminology and Technical Specifications 1st ed.*, Sleep (Rochester). Westchester, IL: American Academy of Sleep Medicine.
- Julius MS, Adi-Japha E (2016) A developmental perspective in learning the mirror-drawing task. *Front Hum Neurosci* 10:1–13.
- Karni A, Sagi D (1991) Where practice makes perfect in texture discrimination: evidence for primary visual cortex plasticity. *Proc Natl Acad Sci U S A* 88:4966–4970.
- Mouze-Amady M, Sockeel P, Leconte P (1986) Modification of REM sleep behavior by REMs contingent auditory stimulation in man. *Physiol Behav* 37:543–548.
- Navarrete M, Schneider J, Ngo H V, Valderrama M, Casson AJ, Lewis PA (2020) Examining the optimal timing for closed-loop auditory stimulation of slow-wave sleep in young and older adults. *Sleep* 43:1–14.
- Plihal W, Born J (1997) Effects of early and late nocturnal sleep on declarative and procedural memory. *J Cogn Neurosci* 9:534–47.
- Porbadnigk AK, Treder MS, Blankertz B, Antons J-N, Schleicher R, Möller S, Curio G, Müller K-R (2013) Single-trial analysis of the neural correlates of speech quality perception. *J Neural Eng* 10:056003.
- Rasch B, Born J (2013) About sleep’s role in memory. *Physiol Rev* 93:681–766.
- Rasch B, Gais S, Born J (2009) Impaired off-line consolidation of motor memories after combined blockade of cholinergic receptors during REM sleep-rich sleep. *Neuropsychopharmacology* 34:1843–1853.

- Riedner B a, Vyazovskiy V V, Huber R, Massimini M, Esser S, Murphy M, Tononi G (2007) Sleep Homeostasis and Cortical Synchronization: III. A High-Density EEG Study of Sleep Slow Waves in Humans. *Sleep* 30:1643–1657.
- Salin-Pascual RJ, Jimenez-Anguiano A, Duran-Vazquez A, Nancy HM, Drucker-Colin R (1994) Administration of auditory stimulation during recovery after REM sleep deprivation. *Sleep* 17:231–235.
- Smith CT, Aubrey JB, Peters KR (2004) Different roles for REM and stage 2 sleep in motor learning: a proposed model. *Psychol Belg* 79–102.
- Snoddy GS (1926) Learning and stability: a psychophysiological analysis of a case of motor learning with clinical applications. *J Appl Psychol* 10:1–36.
- Sockeel P, Mouze-Amady M, Leconte P (1987) Modification of EEG asymmetry induced by auditory biofeedback loop during REM sleep in man. *Int J Psychophysiol* 5:253–260.
- Stickgold R, Whidbee D, Schirmer B, Patel V, Hobson JA (2000) Visual discrimination task improvement: A multi-step process occurring during sleep. *J Cogn Neurosci* 12:246–254.
- Tamaki M, Wang Z, Barnes-Diana T, Guo D, Berard A V, Walsh E, Watanabe T, Sasaki Y (2020) Complementary contributions of non-REM and REM sleep to visual learning. *Nat Neurosci*.
- Valderrama M, Crépon B, Botella-Soler V, Martinerie J, Hasboun D, Alvarado-Rojas C, Baulac M, Adam C, Navarro V, Le Van Quyen M (2012) Human gamma oscillations during slow wave sleep. *PLoS One* 7:e33477.
- Vazquez J, Merchant-Nancy H, Garcia F, Drucker-Colin R (1998) The effects of sensory stimulation on REM sleep duration. *Sleep* 21:138–142.
- Warby SC, Wendt SL, Welinder P, Munk EGS, Carrillo O, Sorensen HBD, Jennum P, Peppard PE, Perona P, Mignot E (2014) Sleep-spindle detection: crowdsourcing and evaluating performance of experts, non-experts and automated methods. *Nat Methods* 11:385–392.
- Widmann A, Schröger E, Maess B (2014) Digital filter design for electrophysiological data - a practical approach. *J Neurosci Methods* 250:34–46.
- Yotsumoto Y, Sasaki Y, Chan P, Vasios CE, Bonmassar G, Ito N, Náñez JE, Shimojo S, Watanabe T (2009) Location-Specific Cortical Activation Changes during Sleep after Training for Perceptual Learning. *Curr Biol* 19:1278–1282.

Reviewers' comments:

Reviewer #1 (Remarks to the Author):

This is an excellent and thorough revision. I have no further comment. Congratulations for this study!

Reviewer #2 (Remarks to the Author):

I thank the authors for responding to my comments, and I'm very glad that my suggestion to correlate the accuracy of the clicks with performance yielded insightful results, with a strong correlation for MTT.

It appears that not all of the new tables presented in the responses to reviewers have been added to the supplement (e.g., I don't see the click accuracy table in the supplement). I would suggest adding all new tables to the supplement in the interest of transparency.

Regarding my concern over the choice of control condition, which the other reviewer shared with me, and the lower accuracy (< 50%) for successfully targeting eye movements: I think the authors have done a satisfactory job addressing this by describing their study as targeting REM in general, rather than targeting individual eye movements (which, as the posthoc accuracy analysis shows, wasn't exactly the case).

Here's my only remaining concern. The authors responded to my comment about cluster permutation statistics by noting that the most common threshold is the t-stat corresponding to $p = 0.05$. That may or may not be true, but I would encourage the authors to consider Smith and Nichols 2009, Neuroimage: "However, a limitation is the need to define the initial cluster-forming threshold (e.g., threshold the raw t-statistic image at $t > 2.5$). This threshold is arbitrary, and yet its exact choice can have a large impact on the results, particularly at the lower (e.g., $t, z < 4$) cluster-forming thresholds frequently used."

Because the authors indeed used a threshold from the lower end, it's advisable to also compare this with a stricter threshold. Also, see the section 'The threshold problem' in Mensen and Khatami 2013 Neuroimage:

"... However, one cannot unequivocally determine which exact threshold will give useful results for a particular dataset, and so any cluster method remains likely to introduce user biases. Rather than using direct t-value cut-offs, some have used uncorrected p-values from the known t-distribution. Although this may give slightly more stable results over different datasets (Hayasaka and Nichols, 2003, Hayasaka and Nichols, 2004, Hayasaka et al., 2004), this represents inconsistent theory in that such a t-value cut-off is calculated based on parametric assumptions, which we have already argued are generally not met in the data. Moreover, even this threshold is arbitrarily selected since there is no theoretical justification as to why a p-value cut-off of 0.05 would produce the most optimal clustering in the dataset. In fact, such a threshold is likely to produce clusters which are too large as this would be the significance threshold for statistics uncorrected for multiple comparisons (Groppe et al., 2011a)."

I completely recognize that it might be very labor-intensive to switch to threshold-free cluster enhancement at this point. Rather than doing so, maybe the authors can simply compare the current clustering with a few other threshold choices, as this could impact the interpretation of results (e.g., the interpretation that one cluster spans the beta frequency range). One principled way to do this

would be to take the absolute value of all t-values in the time-frequency representation and choose the 25th, 50th, and 75th percentiles as thresholds. If the default results using the current threshold look very different from these alternatives thresholds, the authors should address this.

The authors should also specify how they are defining the cluster mass (is it the size of the cluster as measured in pixels, or as the sum of the t-stats within the cluster?).

Minor

1) I take it that the participants weren't asked about dream recall? This might have been an interesting way to infer REM disruption. Have any of the prior studies on auditory stimulation during REM examined its effects on dream recall?

2) There are some grammatical issues, mainly, lots of missing commas. On page 3, the decades should be written either 1980s or '80s (with an apostrophe). Also, in the abstract, "Rapid Eye Movements" doesn't need to be capitalized.

Reviewers' comments:

Reviewer #1 (Remarks to the Author):

This is an excellent and thorough revision. I have no further comment. Congratulations for this study!

We thank the reviewer for his/her feedback and are pleased to know the previous responses were satisfactory.

Reviewer #2 (Remarks to the Author):

I thank the authors for responding to my comments, and I'm very glad that my suggestion to correlate the accuracy of the clicks with performance yielded insightful results, with a strong correlation for MTT.

We thank the reviewer for his/her feedback which made the manuscript better.

It appears that not all of the new tables presented in the responses to reviewers have been added to the supplement (e.g., I don't see the click accuracy table in the supplement). I would suggest adding all new tables to the supplement in the interest of transparency.

We apologize for overlooking this. Previously, we did not include tables when the main points were described in the text. However, acknowledging that some of the values were not discussed but could be of interest to some readers, we now include the new Table S1 (Click stimulation accuracy for REM sleep and EM events.) and Table S7 (Correlations between behavioural tasks and click accuracy (%)). However, for the sake of simplicity, we decided not to include ANOVA tables as these statistics were already included in the text providing enough information about the results.

Regarding my concern over the choice of control condition, which the other reviewer shared with me, and the lower accuracy (< 50%) for successfully targeting eye movements: I think the authors have done a satisfactory job addressing this by describing their study as targeting REM in general, rather than targeting individual eye movements (which, as the posthoc accuracy analysis shows, wasn't exactly the case).

We are glad to read that both reviewers were satisfied with our previous responses.

Here's my only remaining concern. The authors responded to my comment about cluster permutation statistics by noting that the most common threshold is the t-stat corresponding to $p = 0.05$. That may or may not be true, but I would encourage the authors to consider Smith and Nichols 2009, Neuroimage:

“However, a limitation is the need to define the initial cluster-forming threshold (e.g., threshold the raw t-statistic image at $t > 2.5$). This threshold is arbitrary, and yet its exact choice can have a large impact on the results, particularly at the lower (e.g., $t, z < 4$) cluster-forming thresholds frequently used.”

Because the authors indeed used a threshold from the lower end, it’s advisable to also compare this with a stricter threshold. Also, see the section ‘The threshold problem’ in Mensen and Khatami 2013 Neuroimage:

“... However, one cannot unequivocally determine which exact threshold will give useful results for a particular dataset, and so any cluster method remains likely to introduce user biases. Rather than using direct t-value cut-offs, some have used uncorrected p-values from the known t-distribution. Although this may give slightly more stable results over different datasets (Hayasaka and Nichols, 2003, Hayasaka and Nichols, 2004, Hayasaka et al., 2004), this represents inconsistent theory in that such a t-value cut-off is calculated based on parametric assumptions, which we have already argued are generally not met in the data. Moreover, even this threshold is arbitrarily selected since there is no theoretical justification as to why a p-value cut-off of 0.05 would produce the most optimal clustering in the dataset. In fact, such a threshold is likely to produce clusters which are too large as this would be the significance threshold for statistics uncorrected for multiple comparisons (Groppe et al., 2011a).”

I completely recognize that it might be very labor-intensive to switch to threshold-free cluster enhancement at this point. Rather than doing so, maybe the authors can simply compare the current clustering with a few other threshold choices, as this could impact the interpretation of results (e.g., the interpretation that one cluster spans the beta frequency range). One principled way to do this would be to take the absolute value of all t-values in the time-frequency representation and choose the 25th, 50th, and 75th percentiles as thresholds. If the default results using the current threshold look very different from these alternatives thresholds, the authors should address this.

We agree with the reviewer that choosing a threshold value as determined by an arbitrary parametric statistic might lead to generalization to broad clusters built under the spontaneous self-organization of spurious activity. Therefore, as suggested by the reviewer, we analyzed the changes in cluster determination based on several thresholding methods taking as reference the STIM - CNT t statistic as presented in Figure 2b of the original manuscript and shown here in (Figure R1.a).

First, we computed the TFCE clustering using $E = 0.5$ and $H = 2$ (Figure R1.c). This method presented a series of sixteen clusters. The two biggest clusters among them correspond to similar beta and theta clusters as presented in the original cluster analysis thresholding at $p < 0.05$ (Figure R1.b).

Figure R1.

Second, following the reviewer recommendations, we also computed clusters at different percentiles depending on the distribution of absolute t values on the statistic. In this sense, we found very large clusters around the beta and theta activity when evaluated at a t threshold of 25th percentile (uncorrected clusters in Figure R1.d), but none of these survive cluster correction (computed by cluster size). Similarly, a t-threshold of 50th percentile showed large activations at beta and theta (uncorrected clusters in Figure R1.e), but only the beta cluster survived correction. Conversely, when evaluating the thresholding with a t value equivalent to the 75th percentile (uncorrected clusters in Figure R1.f), we found that the beta and theta clusters survived cluster correction. We further evaluated what would be the equivalent alpha

(Figure R2.a) and t-values (Figure R2.b) for 5th and 95th percentiles, and we found that the equivalent t percentile for the selection of our original analysis at $p < 0.05$ is the 73.15th percentile (Figure R2.b). Therefore, the clusters in original analysis are just slightly broader than the clusters at 75th percentile.

Figure R2.

Finally, we wanted to consider how the cluster selection may change when evaluating stricter thresholds. Therefore, we also considered the thresholding using percentiles from the distribution of the permutations for the maximum t statistic. Hence, we evaluated different thresholds from the 1st to the 95th percentile (Figure R3), which reported equivalent alpha values from 0.012 to values lower than 0.00001 (Figure R3.a). Nevertheless, the beta and theta clusters survived across all these different levels of thresholding although with reduced size (Figures R1.g to R1.l).

Figure R3.

Overlapped clusters of interest are displayed in Figure R4, which includes the original analysis with thresholding at $\alpha = 0.05$, the thresholding using TFCE, the thresholding at the 75th percentile of data t-statistic, and the thresholding at the 1st percentile of the permutation of the maximum t-statistic.

Figure R4.

Next, we evaluated how the changes in cluster size will impact the relationships of the spectral response with behavioural changes. We compared thus these relationships using the clusters defined by the 1st percentile of the maximum t-statistic permutation which is stricter than other options while still establishing broad activations across beta and theta frequency bands. Figure R5 replicates here Figures 2c – 2g from the manuscript using this stricter threshold. Hence, when comparing with changes in performance of visual task, we found that a similar fronto-central region is correlated with the beta cluster although the mean activation across channels loses correlation significance with the beta cluster. As in the original analysis, we did not find significant correlations between beta cluster and the procedural task. Instead, when comparing the size-reduced theta cluster with the visual and procedural tasks, we noticed stronger correlations between the average activation across channels and the performance change. Topographically, these activations are more localized in fewer electrodes but still correspond to the same regions as presented in our original analysis.

We can therefore be confident that our main conclusions hold on even after applying stricter thresholds. However, Beta association with visual task seems to be marked by a broader spectrum and expanded time activations whereas theta relationships with performance could be associated with more narrow band activity.

Figure R5.

We now include Figures R4 and R5 in our supplementary data and the main manuscript states:

“Centro-frontal response of both beta and theta clusters (Figure S4) are in keeping with previous studies reporting induced potentials following auditory stimulation in REM sleep²⁸, and they are also evident when using threshold-free or stricter clusters (Figure S5).”

“These results indicate possible interactions between cortical activity and the cognitive memory processes during EMs of REM sleep. Similar trends were observed when applying a stricter cluster threshold (Figure S7). These indicate that activations are more topographically localized, increasing associations between performance with θ_c power while weakening associations between visual performance and relative β_c power.”

We also included in the discussion:

“Our analyses of cortical responses to auditory stimulation showed that changes in EEG spectrum after the click predicted overnight changes in memory performance. Specifically, an increase in post-stimulus Beta power predicted improvements on VDT performance, while a decrease in post-stimulus Theta power predicted a decrease in performance on this improvement on the MTT. Main ERSP activations are also

evident after applying stricter cluster thresholds. However, beta association with visual task seems to be marked by activations that are broader in spectrum and expanded in time, whereas theta relationships with performance were associated with more narrow band activity.”

The authors should also specify how they are defining the cluster mass (is it the size of the cluster as measured in pixels, or as the sum of the t-stats within the cluster?).

We apologize for this information was not clear enough. Cluster size statistic was computed based on the size in pixels of the spectrograms and now this is clarified in the manuscript.

Now the text states:

“Then, the permutation distribution of the maximal suprathreshold cluster size measured in pixels was calculated by re-labelling values of CNT and STM conditions using 1600 non-repeated permutations”.

Minor

1) I take it that the participants weren't asked about dream recall? This might have been an interesting way to infer REM disruption. Have any of the prior studies on auditory stimulation during REM examined its effects on dream recall?

The reviewer makes a good point here, but participants were not asked by any dream recall. We do not have information of previous studies evaluating sound stimulation such as ours and dream associations.

2) There are some grammatical issues, mainly, lots of missing commas. On page 3, the decades should be written either 1980s or '80s (with an apostrophe). Also, in the abstract, “Rapid Eye Movements” doesn't need to be capitalized.

Thank you for noticing this. We have revised the grammar and corrected when required.

REVIEWERS' COMMENTS:

Reviewer #2 (Remarks to the Author):

The authors did a beautiful job analyzing different cluster thresholds. I have no further comments and commend the authors on a great paper.